# In-vivo integration of soft neural probes through high-resolution printing of liquid electronics on the cranium

Young-Geun Park[1,2,8], Yong Won Kwon[1,2,8], Chin Su Koh[3,8], Enji Kim[1,2,8], Dong Ha Lee[1,2], Sumin Kim[1,2], Jongmin Mun[4], Yeon-Mi Hong[1,2], Sanghoon Lee[1,2], Ju-Young Kim[2,5], Jae-Hyun Lee[2,5], Hyun Ho Jung[3]✉, Jinwoo Cheon[2,5,6]✉, Jin Woo Chang[7]✉ & Jang-Ung Park[1,2,3,5]✉

Current soft neural probes are still operated by bulky, rigid electronics mounted to a body, which deteriorate the integrity of the device to biological systems and restrict the free behavior of a subject. We report a soft, conformable neural interface system that can monitor the single-unit activities of neurons with long-term stability. The system implements soft neural probes in the brain, and their subsidiary electronics which are directly printed on the cranial surface. The high-resolution printing of liquid metals forms soft neural probes with a cellular-scale diameter and adaptable lengths. Also, the printing of liquid metal-based circuits and interconnections along the curvature of the cranium enables the conformal integration of electronics to the body, and the cranial circuit delivers neural signals to a smartphone wirelessly. In the in-vivo studies using mice, the system demonstrates long-term recording (33 weeks) of neural activities in arbitrary brain regions. In T-maze behavioral tests, the system shows the behavior-induced activation of neurons in multiple brain regions.

The brain is a three-dimensional (3D), complex network of massive numbers of neurons that constantly generate and transmit signals to communicate, and these neuronal activities and firing patterns control body function, consciousness, and memory formation. The complexity of the brain makes it particularly vulnerable, and subtle abnormalities in neuronal activities can lead to neurological disorders such as epilepsy, Parkinson's disease, Alzheimer's disease, depression, and chronic pain[1]. In this regard, implantable electronic devices that transduce neural ionic signals to electronic signals, i.e., neural probes, have been extensively developed to monitor neuronal activities in a specific region of the brain[2–4]. However, conventional neural probes are based on solid materials, such as metals or silicon, with elastic moduli in the range of hundreds of GPa, which is about seven times of magnitude higher than the elastic modulus of the brain[2,5]. This severe mismatch of mechanical properties causes inflammatory responses of brain tissues and migration of the devices after implantation. Utilizing soft electronics that possess similar mechanical properties as the brain can provide stable contacts with neurons, and recent studies demonstrated that the reduced mechanical mismatch between the electronics and brain elicits a significant reduction of immune responses for chronic implantation[6,7]. Therefore, developing soft and tissue-conformable neural probes can be the key to forming reliable

[1]Department of Materials Science and Engineering, Yonsei University, Seoul 03722, South Korea. [2]Center for Nanomedicine, Institute for Basic Science (IBS), Seoul 03722, South Korea. [3]Department of Neurosurgery, Yonsei University College of Medicine, Seoul 03722, South Korea. [4]Department of Statistics and Data Science, Yonsei University, Seoul 03722, South Korea. [5]Graduate Program of Nano Biomedical Engineering (NanoBME), Advanced Science Institute, Yonsei University, Seoul 03722, South Korea. [6]Department of Chemistry, Yonsei University, Seoul 03722, South Korea. [7]Department of Neurosurgery, Korea University Anam Hospital, Seoul 02841, South Korea. [8]These authors contributed equally: Young-Geun Park, Yong Won Kwon, Chin Su Koh, Enji Kim. ✉e-mail: junghh@yuhs.ac; jcheon@yonsei.ac.kr; jchang@yuhs.ac; jang-ung@yonsei.ac.kr

integration of electronics with the brain, while minimizing injury to the surrounding tissue for chronic neural interfacing.

Furthermore, from the view of the entire system of the neural interface, there are many subsidiary electronics required for collecting and processing the raw signals detected from neural probes[8]. Although numerous studies in developing soft neural probes have improved the long-term stability of the devices and quality of acquired signals, electronics for wireless signal transfer and its electrical interconnections remain as the flat and rigid shape of printed circuit boards (PCBs) based on solid, fragile materials. These structural and material heterogeneities deteriorate the integrity of the neural interface system into biological systems, and restrict the long-term use for subjects during their free moving. In contrast to the flat and rigid PCBs, stretchable designs of electronic devices can be advantageous of being conformally laminated to the curved surface of the body[9–11]. In addition to their stretchability, diversification of the structure design is also important in order to compose device components with various geometries according to the individual subject. For example, the shape and size of the brain and cranium can be diverse for each individual, and also the location of multiple neural probes needs to be different for analyzing and stimulating specific regions of the brain. Various fields of neuroscience research expect to record neural signals from diverse brain regions, and each research requires different placement of multiple neural probes. The multi-region implant is typically enabled by drilling multiple burr holes on the skull and inserting multiple probes, which is a standard process in stereoelectroencephalography (SEEG). Also, other methods, such as the multi-shank neural probes and spatially expandable fiber probes, have been developed to solve these challenges for multiple-region recording[12–14]. However, conventional pre-fabricated and pre-packaged devices with their fixed geometries can limit the spatial variability of neural recordings for each study or subject. In this respect, printing approaches used in the graphic arts, particularly those based on direct writing techniques, are of interest for the flexibility in choice of structure designs, where changes can be made rapidly using software-based printer-control systems. However, the direct printing approach to integrating subsidiary electronics with soft neural probes on the cranium had not previously been demonstrated., and current printable conductive materials that require thermal annealing or solvent-drying processes, which have limited their printability on a biological system[15].

Here, we demonstrate a soft neural probe and their monolithic integration through a neural interface system for the recording of neuronal activities in the brain, using the high-resolution printing of eutectic gallium-indium alloy (EGaIn; 75.5% gallium, 24.5% indium by weight), which is a representative gallium-based liquid metal. The printed neural interface system is composed of soft neural probes which are implanted in the brain, and their subsidiary electronics that are directly printed on the surface of the cranium. This neural interface system includes key advantages as follows. First, soft neural probes are printed of liquid metal with various lengths and a fine diameter of 5 μm. The subcellular-scale diameter of the neural probe is structurally and mechanically similar to neurons and can be implanted in the brain with minimized invasiveness. The length of neural probes can be readily controlled by direct printing of liquid metal through a capillary nozzle, so the neural probes can be precisely adjusted to various depths and locations of the brain. Also, they have good softness comparable to neurons and can restore their electrical conduction against electrical disconnection by possible excessive deformation, due to the low modulus and self-healing capability of liquid metal. Second, the liquid metal-based electrical interconnections and subsidiary electronics are directly printed using a nozzle on the surface of the cranium in vivo. This direct printing of liquid metal on the cranial surface can drastically increase the integrity of electronics to biological systems by forming the circuits and interconnections with conformal

structures along the nonplanar curvature of the cranium. Using the ability to form the miniaturized and conformal circuits on cranium directly, we demonstrate a cranial circuit for wireless neural recording of a live mouse. Also, the liquid metal-based cranial electronics instantly operate after being printed without additional post annealing processing. Third, the soft neural probes and cranial electronics are integrated as neural interface systems with adaptable geometries. Our soft neural probes are implanted in multiple brain regions of a single subject, and the subsidiary circuits and interconnections are adaptably printed to the configuration of multiple neural probes directly on the cranium, allowing the structural design of neural interface systems to be diversified. Furthermore, the in-vivo characterization of wireless neural recording and behavioral tests using freely moving mice demonstrate the simultaneous recording of local field potentials (LFPs) and single-unit spikes in multiple regions of the brain through the printed neural interface system, as well as its biocompatibility and stability for a long-term signal recording (33 weeks).

## Results

### Printing of soft neural probes

For monolithic integration of soft neural probes and their subsidiary electronics, we designed a soft neural probe with liquid metal through a high-resolution direct printing. Figure 1a illustrates the printing system of liquid metal[16]. This system consists of a nozzle connected to an ink reservoir, a pneumatic pressure controller which controls the pressure to deliver the ink from a reservoir onto the substrate, and a 6-axis stage with automatic movements in the x, y, or z-axis, two tilting axes and one rotating axis in the xy-plane. EGaIn was used as an ink. To print EGaIn, we prepared the nozzles using a pipette puller to make glass capillaries with inner diameters of 5 to 60 μm, or plastic nozzles with metal tips (NNC-PN, NanoNC, Republic of Korea) were used with inner diameters of 120 and 150 μm. Using a nozzle with an inner diameter of 5 μm, the distance between the nozzle tip and the substrate was controlled to be 2 μm, and the pneumatic pressure of 50 psi was applied to deliver the ink (i.e. EGaIn) from a reservoir onto the substrate through a nozzle. EGaIn was directly printed by coordinating the operation of this pneumatic pressure (on/off) and the movement of translation stages. Figure 1b shows the photograph and scanning-electron micrograph (SEM) of liquid metal patterns with a width of 5 μm and variable lengths, for the electrode of soft neural probes. This facile control of pattern length can flexibly form neural probes with various lengths. Also, the inner diameter of the nozzle can control the printed line-width of EGaIn. Figure 1c shows the relationship between the inner diameter of a nozzle and the line-width for a constant printing velocity of 0.1 mm/s. As the diameter of the nozzle increased, the printed line-width became increased almost linearly. After the liquid metal lines with 5 μm in the width were printed for neural probes, parylene was deposited to the entire surface for the passivation of liquid metal electrodes, except the tip part of the printed EGaIn. The detailed fabrication process of the soft neural probes is presented in the Methods section. Figure 1d shows the neuron-like dimension of our soft neural probe. The soft neural probes had an electrode of 5-μm in line-width, and it is comparable to the widths of axons that carry action potentials[3]. The opening of the electrode for tissue interfacing was 5 μm in width and 7.5 μm in height (Fig. 1e). This fine area of our liquid-metal electrode (the overall surface area of this exposed cylinder: 58.8 μm²) results in high spatial resolution for single-neuron-scale recording.

To further enhance the signal quality of this small opening of an electrode, we deposited platinum nanoclusters, denoted as platinum black (PtB)[17], only at the open area of this liquid-metal electrode by electrodeposition (Fig. 1f). This deposition of PtB resulted in significantly lower impedance due to the increase of electrochemical surface area by nanostructure. Figure 1g presents the impedance spectroscopy of PtB-coated liquid metal (PtB/EGaIn) and pristine

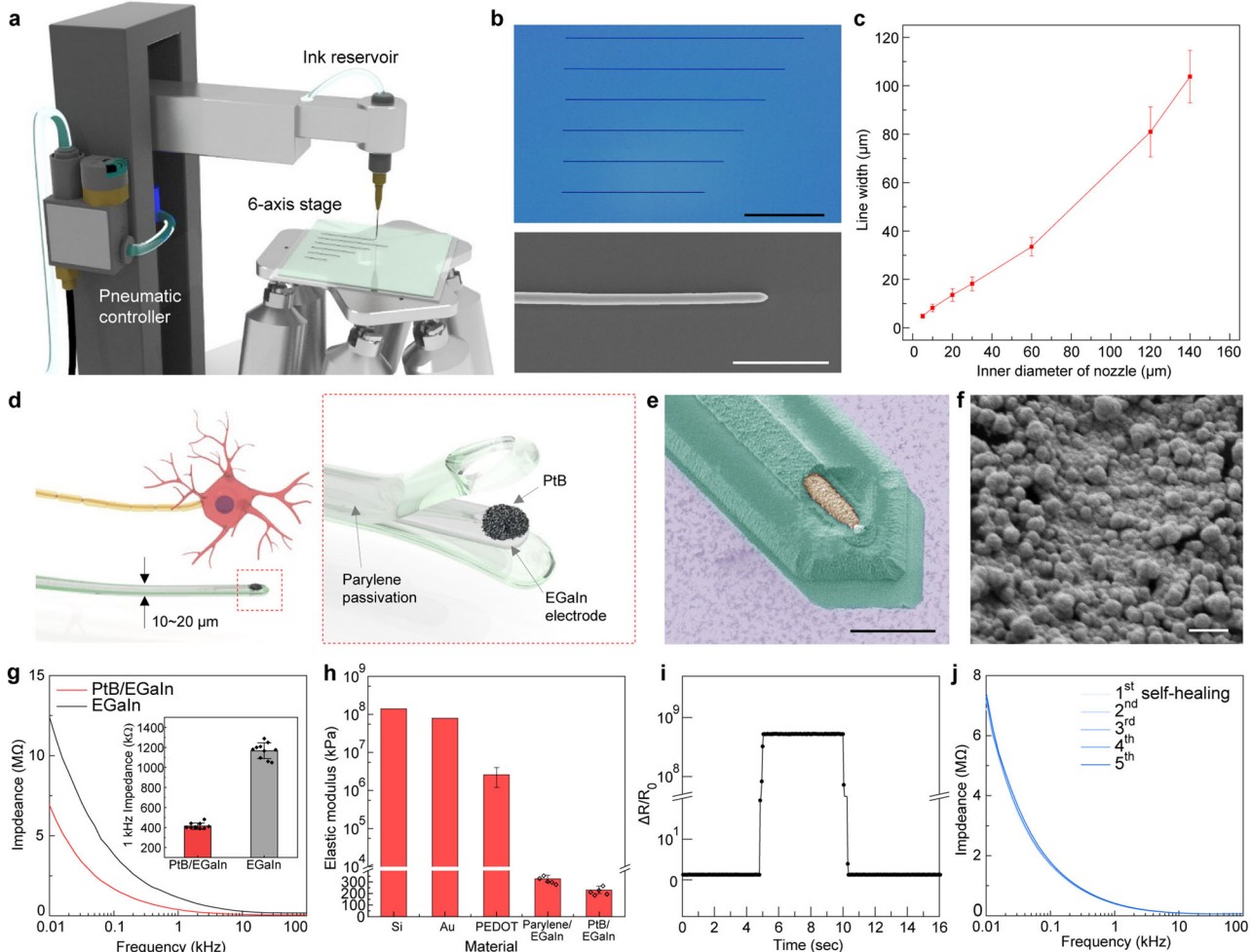

**Fig. 1 | Printing of soft neural probes. a** Schematic illustration of a printing system of liquid metal. **b** Photograph and SEM of printed liquid metal patterns with various lengths. Scale bars, 500 μm (black) and 50 μm (white). **c** Line-widths of printed liquid metal as a function of an inner diameter of the nozzle. Each data point indicates the average of five measurements, and error bars represent the standard deviation. **d** Schematic illustration of a structural similarity between the soft neural probe and neuron. Inset: the schematic illustration of exploded view of soft neural probe. **e** Colorized SEM of a tip part of the soft neural probe. Green and yellow colors correspond to parylene and PtB, respectively. Scale bar, 10 μm. **f** SEM of the

PtB deposited on the tip part of the soft neural probe. Scale bar, 500 nm. **g** Impedance spectroscopy of pristine EGaIn and PtB-coated EGaIn. Inset: Impedance values of PtB/EGaIn and EGaIn at 1 kHz. Error bars represent the standard deviation ($n = 10$). **h** Comparison of elastic moduli of silicon, gold, PEDOT, Parylene/ EGaIn, and PtB/EGaIn. Error bars represent the standard deviation ($n = 5$ independent samples). **i** Real-time change in resistance of a PtB/EGaIn during its disconnection and coalescence. **j** Impedance spectroscopy of PtB/EGaIn during repeated disconnection and coalescence.

EGaIn. PtB/EGaIn showed an impedance of 415.0 kΩ at 1 kHz sampling, which was about 3 times lower than that of the pristine EGaIn (1167.3 kΩ at 1 kHz).

Neural recordings can be influenced by a variety of biological factors, such as blood oxygenation in the tissue, shifts in expression levels of vesicular transporter proteins and ion channels, axon and myelin injury, and interrupted blood flow at the device/tissue interface[18]. Also, the softness of neural probes can be considered one of the main factors that enable a compatible interface with the brain tissue by minimizing inflammation and scar information. We measured the elastic moduli (E) of parylene-coated EGaIn (Parylene/EGaIn) and PtB/EGaIn by tensile tests, and the elastic moduli were compared to that of conventional materials for neural electrodes (Fig. 1h). Although there was a formation of thin layer of PtB on EGaIn, the elastic modulus was negligibly changed from that of the pristine EGaIn. The elastic moduli of Parylene/EGaIn and PtB/EGaIn was measured as 330 kPa and 233 kPa, respectively, and these values were about six orders of magnitude lower than that of solid-type silicon and the gold, and even four orders of magnitude lower than poly(3,4-ethylene dioxythiophene) (PEDOT) ($E = 2.6 \times 10^6$ kPa)[19,20]. In contrast to the rigid probes that use

solid materials, this extraordinary softness of our neural probes can be advantageous to minimize the inflammation for long-term neural interfacing. Especially, the low modulus of Parylene/EGaIn can be further reduced by decreasing the parylene thickness during the probe fabrication process. The bending stiffness of our soft neural probe can be calculated using the modulus value of Parylene/EGaIn[21], and was approximately 0.264 pN·m. This stiffness is comparable to that of brain tissue (0.001–1 pN·m)[22].

The reliable operation of neural probes can be limited by the failure of the device, possibly due to the accumulated mechanical stresses by subtle movements of a subject or excessive deformation during the implantation process. In these situations, liquid metal can work as self-healable electrodes to overcome the vulnerability of thin, soft electronics by recovering their electrical conductance after physical disconnections of conductive pathways. Figure 1i shows the real-time change in electrical resistance of PtB/EGaIn. The PtB/EGaIn was split off by excessive stretching of this sample with a tensile strain of 250% around 5 s, which increased the resistance significantly to its electrical disconnection. Additionally, the resistance recovery to its original status by the coalescence of this sample occurred very rapidly

at ambient conditions (within around 10 s). As the neural probe needs to operate in a wet condition of brain tissue, self-healing of electrochemical properties was tested in a saline solution by impedance measurement. After 5 times repetition of its disconnection and reconnection, this PtB/EGaIn sample had a negligible change in impedance spectrum (Fig. 1j).

## Printing of conformal electronics on a cranial surface

Printed electronics has been rapidly evolving over the past few decades, and represents huge potential to enable biomedical devices and electronics to be fabricated directly on the biological tissues. In biomedical applications, the target live biological surfaces are very vulnerable to dehydration and heat. However, most of the printable conductive inks for electronic devices require drying or thermal annealing processes. Although hydrogel-based conductors can be printed on biological surfaces, their low electrical conductivities can limit their use in state-of-the-art, high-performance electronics. In contrast, gallium-based liquid metals can offer high conductivity with no additional processes, so the damage to the biological surface can be minimized. Also, some latest studies of these liquid metals also revealed their negligible toxicity and biocompatibility[23].

Figure 2a and Supplementary Movie 1 show a conformal 3D printing of EGaIn directly on the cranium of a living mouse. Based on the direct printing system in Fig. 1a, a rodent adapter with ear bars and jaw cuff (RWD Life Science, USA) was equipped on the 6-axis movement stage. Before the printing of cranial electronics on the cranium, the medical-grade polymer (Loctite 4011, Henkel, Germany) was printed along the curvature of the cranium for passivation. Curing time of this polymer layer depends on its thickness, and can be significantly shortened as this layer becomes thinner. Supplementary Fig. 1 presents the sweeping test result of a 30-μm-thick Loctite 4011 layer, and the 2-min curing time allowed its mechanical hardening for subsequent processing to form our cranial electronics. Furthermore, the cytotoxicity of this Loctite 4011 layer (thickness: 30 μm, curing time: 2 min) was evaluated by Live/Dead assay (Supplementary Fig. 2). An anesthetized mouse (C3H, male, 14 weeks) was fixed on the adapter to minimize the subtle motion or vibration from heartbeat and breathing. During printing, the distance between the nozzle tip and the cranial surface was controlled to be ~5 μm, and constant pneumatic pressure of 60 psi was applied. Figure 2b, c present optical micrographs of the liquid-metal pattern printed with complex geometries on the cranial surface of a mouse. This high-resolution, conformal pattern of liquid metal along the curvature of the cranium was printed with the minimum line-width of 5 μm. The printing velocity was 80 μm/s and the total printing duration was 22 min, which was within the anesthesia time of a mouse.

Figure 2d, e shows a SEM and an optical micrograph of the near-field communication (NFC) chip and chip antenna interconnected by printing free-standing 3D structures of EGaIn directly on a cranial surface of a mouse. Here, a medical-grade silicone adhesive (Kwik-Sil, World Precision Instruments, USA) was used to mount these chips on the cranium before printing, and then multiple interconnection lines of EGaIn were printed with 3D structures. The diameter of these 3D interconnections was ~5 μm, which is smaller than the resolution of current wire-bonding technologies (wire diameter > ~20 μm) in conventional electronics. The printing velocity was 50 μm/s in average and the total printing duration was 12 min. These liquid- metal interconnections were formed to contact with metal pads of the NFC chip and the chip antenna directly on the cranium, which made the additional soldering process unnecessary. This printed circuit was additionally encapsulated by covering a medical-grade silicone layer to prevent electrical disconnections of interconnections during the skin closure and suture process. The silicone layer was cured within 15 min at room temperature, so the total process time for fabricating the wireless printing system in the mouse was 28 min, which was also

within the anesthesia time of a mouse. During the encapsulation and skin-suture processes, high elasticity of these EGaIn interconnections minimized their electrical disconnections with the NFC chip. Supplementary Fig. 3 shows the layout of the circuit schematically. Using the circuit, we tested the wireless communication between the NFC chip and a smartphone, across the scalp skin (Supplementary Fig. 4 and Supplementary Movie 2). Within the distance of ~2 cm between the smartphone and this cranial circuit (located below the scalp), two antennas were coupled and the information in the NFC chip was able to be transmitted to the smartphone.

Similarly, deep brain neural signals recorded by our soft neural probes (presented in Fig. 1) can be wirelessly transferred to a smartphone by printing a circuit on the cranium of a live mouse. The block diagram in Fig. 2f shows the system architecture of this cranial circuit for wireless neural recording. It included two soft neural probes (each for neural recording and reference electrode), an analog front-end (AFE) chip circuit for filtering and biosignal amplification (ADS1201, Texas Instrument, USA), an NFC microcontroller unit (MCU) chip (RF430FRL152H, Texas Instrument, USA), and a chip-type antenna for power and data transfer (W3102, Pulse Electronics, USA). The 14-bit analog-to-digital converter (ADC) in NFC MCU digitized the recorded LFP signals at a sampling rate of 153 Hz. The NFC-compatible smartphone wirelessly delivered the power for operating the chips by providing the AC bias to the chip-type antenna. These probes were implanted in hippocampal region of a mouse brain, and all electronic components and the neural probes were interconnected on the cranium by printing EGaIn, after coating a thin and transparent insulating layer (Loctite 4011, Henkel, Germany) on the cranium to mount these chips. Then the encapsulation using the medical-grade silicone as well as subsequent skin closure and suture steps completed the overall process. The total process time for fabricating the wireless printing system in the mouse head was 23 min. For the acclimatization of mouse, neural recording was demonstrated 7 days after the implantation of neural probes and the formation of cranial circuits. Figure 2g and Supplementary Movie 3 show the NFC-based wireless neural signal recording of a live, anesthetized mouse using a smartphone. This wireless recording was performed 7 days after the probe insertion and circuit formation to ensure the mouse acclimatization. With the distance of ~15 mm between a smartphone and the mouse head, LFP signals in hippocampal region were clearly detected and transferred to the smartphone wirelessly (Fig. 2h). Additional wireless LFP signals from the other two mice are plotted in Supplementary Fig. 5. After the anesthesia wore off, the mouse showed its healthy movement.

To ensure the stability of the cranial circuit in mice, the circuit was visualized using micro-computerized tomography (micro-CT) 6 weeks after the circuit formation. As shown in Supplementary Fig. 6, the circuit printed between the encapsulation layers of silicone elastomer exhibited no disconnection or degradation of EGaIn traces. The overall photographs of cranial circuits and its corresponding circuit diagrams are shown in Supplementary Fig. 7. Additionally, we tested the mechanical stability, long-term reliability, and signal quality of the NFC-based wireless cranial circuit. To test the mechanical stability, an NFC-based wireless circuit was formed on a curved structure using our direct printing method (Supplementary Fig. 8a), and evaluated the extent of circuit deformation under a compressive force of 1 N was applied to the sample (Supplementary Fig. 8b). This is approximately 3 times the weight of an adult mouse (~0.3 N), in consideration of the impact that may occur during the normal behavior or group living of subjects. Magnified stereomicrographs taken before and after application of this force showed that the 3D interconnects did not collapse and the circuitry was not damaged, due to the elastomeric encapsulation layer (Supplementary Fig. 8c). To verify the long-term reliability, we fabricated an NFC-based wireless circuit on the cranium of a mouse, specifically the open pads of the NFC chip and antenna were

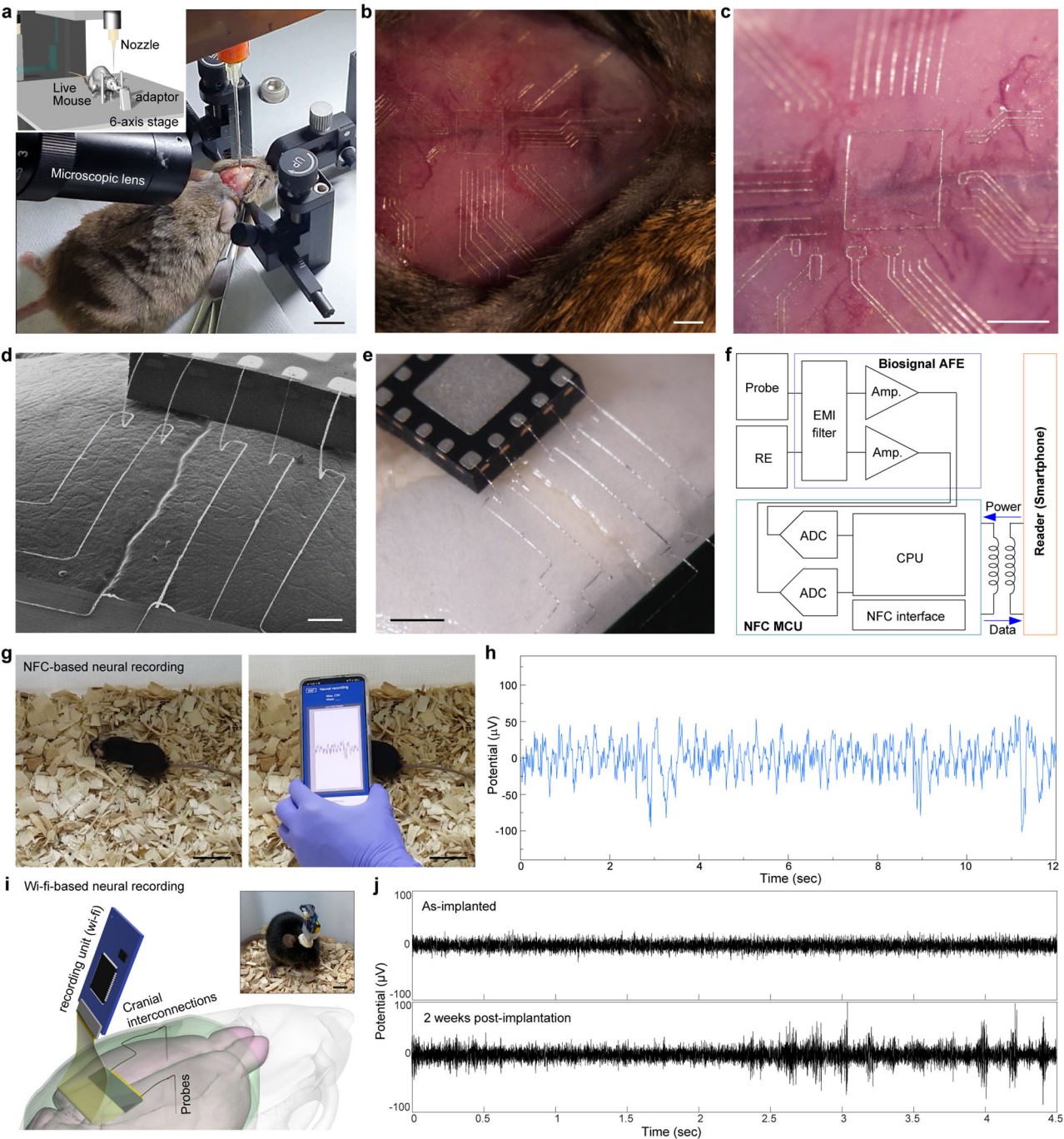

**Fig. 2 | Printing of liquid metal circuits on cranial surface. a** Photograph showing the printing of liquid metals on the cranial surface of a mouse. Scale bar, 1 cm. Inset: schematic illustration showing the setup of conformal printing. **b** Optical stereomicrograph of liquid-metal patterns printed on the cranial surface of a mouse. Scale bar, 1 mm. **c** Magnified view of optical stereomicrograph of liquid-metal patterns printed on the cranial surface of a mouse. Scale bar, 1 mm. **d** SEM of wireless cranial circuit including NFC chip, liquid-metal interconnection, and a chip antenna. Scale bar, 500 μm. **e** Optical stereomicrograph of the wireless cranial circuit. Scale bar, 500 μm. **f** Block diagram of a cranial circuit for NFC-based wireless neural

recording. **g** Photographs of the NFC-based wireless neural recording in mouse brain through smartphone. Scale bar, 4 cm. **h** Representative LFP trace recorded and wirelessly transferred by NFC-based neural recording system with a cranial circuit. **i** Schematic illustration of a long-range wi-fi neural recording system in the mouse brain. Inset: photograph of the mouse with this wi-fi neural recording system. Scale bar, 2 cm. **j** Single-unit traces recorded using this integration of a wi-fi recording unit and cranial circuits for the as-implanted case (top) and the 2 weeks post-implantation (bottom).

electrically connected by printing EGaIn interconnects prior to their encapsulation using an elastomer, and this resulting sample was then subjected to the accelerated aging test in an in-vitro environment (75 °C and 7 days, a day is equivalent to a month in the normal environment). The relative change in electrical resistance of this sample was negligible (Supplementary Fig. 9).

We compared the signal recording quality of our monolithic connection (EGaIn neural probe – EGaIn interconnect) to heterogeneous connection (EGaIn neural probe – Au interconnect) through an in-vitro setup that mimicked the signal recording process from the brain to analyzer via a neural probe and an interconnect (Supplementary Fig. 10a). Here, our soft neural probes were connected to

evaporated Au or printed EGaIn interconnects (line-width: 5 μm, pitch: 100 μm), respectively. These neural probe-interconnect pairs were then assessed for signal quality by transmitting the controlled electrical signals (i.e. biphasic current pulse of 0.5 mA amplitude, 0.2 ms pulse width, and 0.02 ms interphase interval) to each pair connected with a potentiostat. (Here, biphasic current pulses were transmitted to generate the conditions similar to those for signals to pass through brain tissue to our probes and interconnections.) As shown in Supplementary Fig. 10b, the monolithic pair acquired 1.5 times higher potentials compared to the heterogeneous pair, possibly due to the minimal contact resistance between the monolithic EGaIn neural probe and EGaIn interconnect. Also, our monolithic pair recorded potentials with less distortion of the injected pulse waveform. In addition, there was no significant influence on the signal waveform and delay between two interconnections in both cases, indicating negligible noise coupling. We also compared the signal quality of single-unit spikes in mouse brains using (i) our soft neural probes with our cranial electronics, as well as (ii) conventional Nichrome electrodes with pre-designed PCB connectors (Supplementary Fig. 11). Both systems were each interfaced with a wi-fi module (JAGA Penny, Jinga-hi, USA). After obtaining signals from each probe, we calculated the signal-to-noise ratio (SNR), the ratio of root-mean-squares (rms) of signals to that of noises. For the conventional system, the SNR was calculated to be 3.483 in average. In comparison, our system's SNR was 5.977 in average, indicating approximately 1.5 times improvement in signal quality compared to the conventional system.

The adaptability of direct printing effectively addresses individual variances in the shape and size of the brain and cranium, allowing customizable circuit configurations for specific needs. For example, soft neural probes can be implanted in multiple brain regions to target a variety of diseases that vary among individuals, with corresponding subsidiary circuits and interconnections that can be adaptively printed. For example, Supplementary Fig. 12 presents four different configurations of cranial circuits to demonstrate the capability of analyzing multiple regions within a brain using customizable circuits.

For the practical usability of neural recording, a long-range wi-fi neural recording unit was also demonstrated (Fig. 2i). A commercial wi-fi neural recording unit (JAGA Penny, Jinga-hi, USA, size: 1 × 3 cm) was interconnected to our neural probes by the liquid-metal printed cranial circuit. Single-unit traces of the as-implanted and 2 weeks post-implantation were acquired through the wi-fi wireless recording system at a sampling rate of 20,000 Hz. Compared to the signal from the as-implanted case (Fig. 2j (top)), the signal amplitude increased 2 weeks after the implantation (Fig. 2j (bottom)). This wi-fi neural interface system negligibly caused acute immune response to the brain, enabling stable recording of neural activities. This wi-fi-based wireless recording was tested in three different mice (Supplementary Fig. 13). To evaluate the working distance of the wireless recording system, a mouse with this wireless neural interface was placed (inside a cage) in an open space without other objects. In this environment, stable neural signals were acquired 2 meters away from the mouse at a sampling rate of 20,000 Hz.

## Monolithic integration of neural interface system

By combining the implantable soft neural probes and high-resolution printing of liquid metal on a cranial surface, a monolithic liquid metal-based neural interface system can be formed. Figure 3a illustrates a layout of the neural interface system. First, a thin and transparent passivation layer was printed on the cranial surface to prevent the potential electrical leakage through the wet cranium. Second, the liquid-metal based, soft neural probe was loaded in a pulled glass capillary filled with suspending solution (PBS), and then stereotaxically implanted to a specific brain region through predrilled holes in the cranial bone using a pulled capillary needle[3,24] with inner and outer diameters of 50 and 70 μm, respectively. This implanted glass capillary

was then slowly pulled away from the brain tissue at a steady rate. Simultaneously, the fixed volume of PBS and suspended probe was expelled out of the capillary using a syringe pump. By matching the retraction velocity and volumetric flow rate, the soft neural probe remained in its initial position during the removal of this glass capillary[15] (Supplementary Fig. 14). As neural probes could be formed with adaptable lengths to a specific depth of brain region by printing, the outer end could be controlled to be exposed on the cranial surface (Supplementary Fig. 15). The optical micrograph of Supplementary Fig. 16 presents that the implanted neural probes maintained their straightened shape without bending inside a brain tissue. Third, the liquid metal-filled printing nozzle contacted the exposed tip of the neural probe on the cranium, and the interconnections of liquid metal were printed directly on the cranium area for electrically connecting multiple neural probes to signal processing units with arbitrary positions. Finally, the encapsulation layer of a silicone-based medical adhesive was deposited on the interconnected neural interface system before suturing the scalp layer to cover this encapsulation. The integrated neural interface system is shown in Fig. 3b, and overall processes are schematically illustrated in Supplementary Fig. 17. Total 12 neural probes were implanted in motor cortices (MO), hippocampal regions (HIP), and visual cortices (VIS) of both hemispheres of a live mouse brain, and these probes were interconnected through the cranial interconnections to the signal processing units. The printing velocity was 60 μm/s in average and the total printing duration was 19 min. Figure 3c illustrates the top-view of the printed neural interface system in Fig. 3b, and indicates the relative position of neural probes (as black dots) and their cranial interconnections.

The soft neural probes exhibited low impedance of 412.0 kΩ at 1 kHz on average, with a standard deviation of 13.3 kΩ, presenting the good uniformity in impedance of multiple neural probes (Fig. 3d). Also, this impedance measurement through the cranial interconnections demonstrated reliable electrical contacts of the printed cranial interconnections with the implanted neural probes. Furthermore, the probes were connected to the wi-fi wireless recording unit using our cranial printing method, and simultaneous 12-channel recording of single-unit spikes exhibited stable detections of neural signals spanning the multiple brain regions. Figure 3e, f shows the recorded single-unit traces and principal component analysis (PCA)-clustered single-unit spikes at 16 weeks post-implantation, respectively. As plotted in Fig. 3e, all of the 12 neural probes detected the single-unit traces which exhibited comparable, low noise levels (~15 μV). The mean waveforms of single-unit spikes recorded by 12 neural probes are presented in Fig. 3f. In addition, the propagation of neural activity was detected using this neural probe array. For example, Fig. 3g, h plot hippocampal LFPs and their superimposed theta waves (4–8 Hz oscillation) recorded in channels 9 and 10. These two LFP traces exhibited similar rhythms in amplitude as shown in Fig. 3g, and the temporal delay shown in superimposed theta waves represented the propagation of field potentials from channel 9 to 10 (Fig. 3h)[25].

Figure 3i shows the time-dependent change in amplitude of the representative single-unit spike (channel 10) during 33 weeks of chronic recording. At one week after implantation, the mean amplitude of single-unit spikes recorded by 12 neural probes had a peak-to-peak amplitude of ~24 μV. Upon the probe insertion, acute insertion injury could reduce the neuronal cell density around the probe, and therefore reduced signal amplitude during the first five weeks[26]. And then this amplitude gradually increased to reach ~143 μV at 15 weeks post-implantation, possibly as the tissue recovered from the acute damage[3]. After 25 weeks, the amplitude of spikes slightly reduced to reach ~125 μV at 33 weeks post-implantation. In the first week of probe injection, 42% of channels (5 of 12) detected sortable single-units spikes. However, from 2 weeks post-implantation, the single-unit recording yield increased to 66% and reached 100% at 11 weeks post-implantation with detection of a total of 18 sortable single-unit spikes.

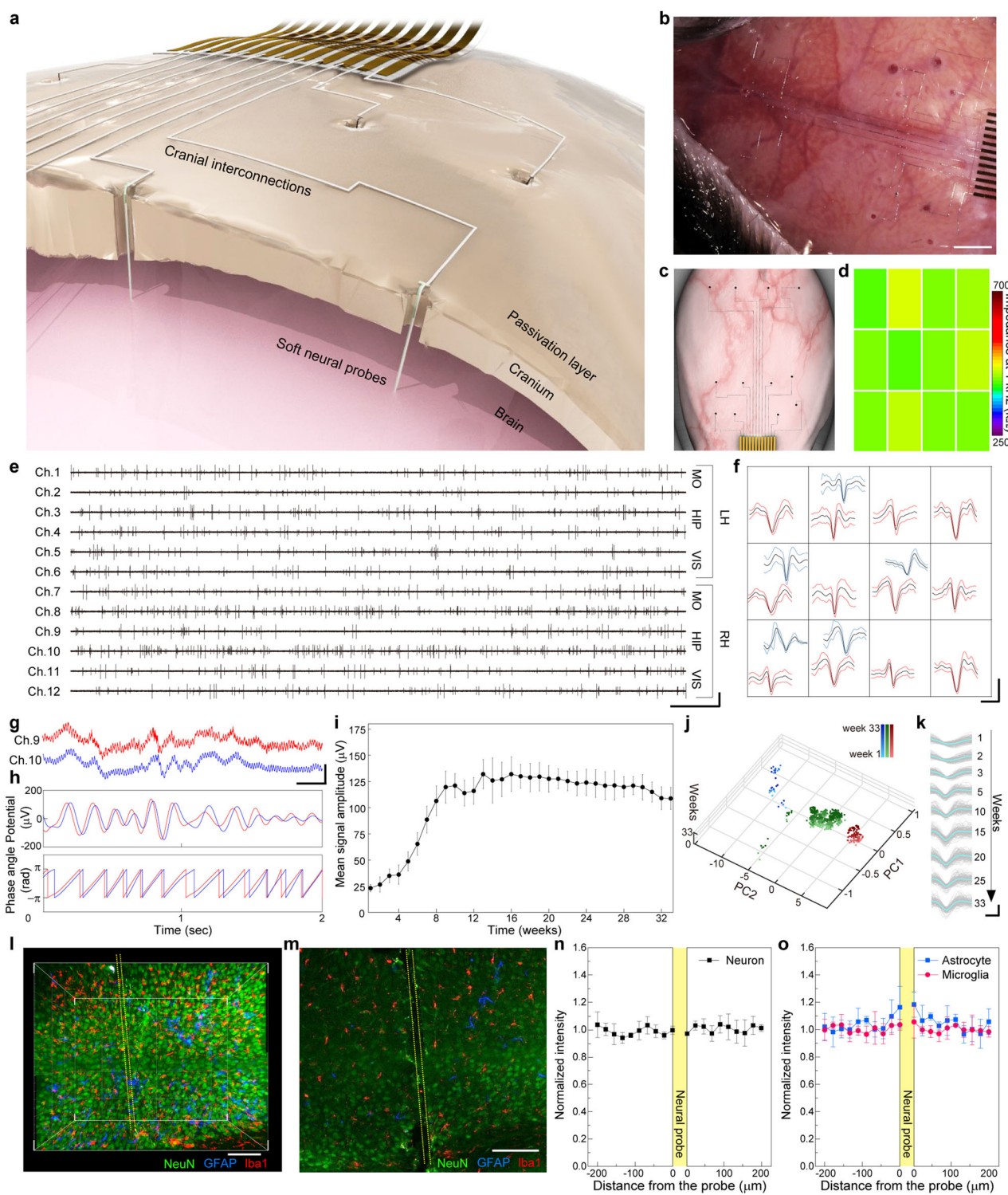

As shown in Supplementary Fig. 18, this high yield of single-unit recording maintained until the end of the long-term recording (up to 33 weeks). In order to confirm the stability in this long-term recording, we carried out PCA of the recorded single-unit spikes for 33 weeks (Fig. 3j, k). The sorted spikes indicated nearly constant positions in each principal component plane during this long-term recording (Fig. 3j). Time-evolution plots of PCA-clustered single-unit spikes in channels 2, 5, 7, and 9 are presented in Supplementary Fig. 19, with the calculation of isolation distances and *L*-ratios of each cluster. Also, Fig. 3k represents the similarity of waveforms (green clusters in Fig. 3j). These results can suggest that the position of the implanted neural

probe was negligibly moved from its initial location with recording the identical neuron during the measured period, and the neuron maintained its viability of being adjacent to the neural probe. Supplementary Fig. 20 shows time evolution of spikes normalized to the maximum potential, to show the similarity of relative waveforms. The isolated neurons recorded in the channel showed constant waveforms during the long-term recording, which represents the stable recording of the identical neuron.

Figure 3l presents a representative 3D reconstructed micrograph of the hippocampal region (where a soft neural probe was pre-implanted) after 33 weeks post-implantation. Visualization of cell types

**Fig. 3 | Monolithic in-vivo integration of neural interface system. a** Schematic illustration of neural interface system. **b** Optical stereomicrograph of neural interface system applied to a mouse. Scale bar, 1 mm. **c** Schematic illustrations showing the position of neural probes and the cranial interconnections. **d** Impedance map of implanted electrode arrays. **e** Single-unit traces of 12-channel neural probe arrays in mouse brain after 16 weeks post-implantation. MO motor cortex, HIP hippocampus, VIS visual cortex, LH and RH left and right hemisphere, respectively. Scale bars, 200 μV (vertical) and 500 ms (horizontal). **f** PCA-clustered single-unit spikes for each channel. Scale bars, 100 μV (vertical) and 2 ms (horizontal). **g** LFPs recorded from hippocampus in RH by channel 9 (red) and 10 (blue). Scale bars, 500 μV (vertical) and 200 ms (horizontal). **h** Superimposed theta waves and corresponding theta angles extracted from LFPs of (**g**). **i** Mean signal amplitudes of single-unit spikes recorded at 12 soft neural probes for 33 weeks of chronic

recording. Each data point indicates the average of ten measurements, and error bars represent the standard deviation. **j** Time evolution of PCA-clustered single-unit spikes from channel 10 over 33 weeks after injection. **k** Single-unit spikes recorded at a channel 10 for 33 weeks of chronic recording. Scale bars, 100 μV (vertical) and 0.5 ms (horizontal). 3D-reconstructed confocal micrograph (**l**) and fluorescence micrograph (**m**) showing implanted neural probes in hippocampal region. Yellow lines indicate the boundary of the implanted neural probe. Scale bars, 100 μm. Normalized fluorescence intensity of neurons (**n**), astrocyte and microglia (**o**) as a function of the distance from the boundaries of the implanted neural probe ($n = 6$). Each data point indicates the average of five measurements from each probe-implanted tissue taken from six mice, and error bars represent the standard deviation.

was assessed by staining neurons, astrocytes, and microglia with NeuN protein (anti-FOX3), glial fibrillary acidic protein (GFAP), and allograft inflammatory factor 1 (Iba1) antibody, following the tissue clearing process (See Methods section)[27]. Figure 3m shows a magnified confocal micrograph near the neural probe location in hippocampal region of Fig. 3l (color-blind safe images are presented in Supplementary Fig. 21). The soft neural probe was implanted with its straightened shape without crumpling or bending (Supplementary Fig. 16), which can be advantageous for the precise, targeted delivery of soft neural probes into specific stereotaxic coordination. Also, quantification of fluorescence intensities of neurons, astrocytes, and microglia was performed using tissue slices from six different mice 8 weeks after neural probe implantation. The interface between the neural probe and brain tissue showed neither significant depletion of neurons (Fig. 3n), nor the enhancement of astrocyte and microglia (Fig. 3o), with presenting relatively uniform cell distribution near the probe.

To evaluate cytotoxicity, we cultured human neuroblastoma SH-SY5Y cells in media pre-contained with pristine EGaIn and PtB/EGaIn samples. After seven days, we performed the MTT and Calcein assays. As shown in Supplementary Fig. 22, both cultures (in media pre-contained with these pristine EGaIn and PtB/EGaIn samples) showed reliable viability comparable to the control sample, indicating the negligible toxicity. To test the immune response of brain tissue to our neural probe, an immunohistochemical assay was implemented for astrocytes and microglia staining 8 weeks after the implantation. As shown in Supplementary Fig. 23, there were negligible differences in the density of neurons as well as those of astrocytes and microglia, indicators of immune response. These results indicate that utilization of our soft neural probes did not significantly elicit glial responses. Also, the scalp of a mouse with the cranial circuit was characterized by H&E staining 6 weeks after this circuit formation (Supplementary Fig. 24). The histology image shows that the scalp in direct contact with the cranial circuit and encapsulation showed neither malignancy nor inflammation.

### Behavioral studies using the soft neural interface system
The neuronal activities of displace units in the hippocampal region show dramatic changes in firing rate during active movements of a subject such as its running, rearing, and exploratory sniffing[28–30]. Also, neurons of the visual cortex in rodents are affected by various receptive fields[31]. Especially, layer 6 (L6) principal neurons in the mouse primary visual cortex associate with the head-rotation motion and visual-based information[32]. To demonstrate the detection of neuronal activities of a freely behaving mouse using our neural interface system (i.e., soft neural probes in the brain and the subsidiary circuit on a cranial surface), two neural probes were implanted in the hippocampus and visual cortex (Fig. 4a) with their interconnection to a wi-fi wireless neural recording unit. Behavior tests with the electrophysiological recording of a freely moving mouse were implemented using a T-shaped maze (T-maze) to observe the activation of displace

units in the Cornu Ammonis 1 (CA1) field of the hippocampus and L6 principal neurons in the visual cortex, and four different mice were tested to ensure the statistical significance. This T-maze was an enclosed apparatus in the form of a T placed horizontally, consisting of one start arm and two goal arms. We added patterns in each goal arm, which could work as visual cues for the additional neuronal and behavioral responses driven by visual-based information (Supplementary Fig. 25). T-maze can be used in a variety of ways to assess the cognitive ability of an animal by promoting the behavioral strategies of rodents in navigation-based tasks[33,34]. We targeted the activation of neural signals by two possible tasks of mice in T-maze (Fig. 4b). The first task was the active locomotion of mice, and this task could activate displace units in the hippocampus. The second task was the head rotation of the mouse at the start arm-goal arm junction, and this task could affect the neuronal activity in the primary visual cortex. The position and movement of the mouse were tracked during its freely moving condition in T-maze (Fig. 4c and Supplementary Movie 4), by the real-time recording of neuronal activities in CA1 and L6 regions (Fig. 4d). When the mouse moved rapidly forward ($0 < t < 1.8$ s), the firing rate of single-unit spikes in the CA1 noticeably increased, as shown in the spike raster of Fig. 4d. When this mouse stopped its active locomotion after $t = 1.8$ s, however, the firing rate of hippocampal neuronal spikes significantly reduced. During the head-rotating motion of this mouse ($1.2 < t < 3$ s) to the left goal arm, the marked increment of neuronal activities in primary visual cortex L6 was detected. Results from three other mice are presented in Supplementary Fig. 26.

The hippocampal neuronal activities were monitored during the running state (Fig. 4e–g) and standstill state (Fig. 4h–j) of this mouse. Large fluctuation in LFPs was recorded during the displacement behaviors (i.e., running) (Fig. 4e), as opposite to the weak fluctuation in LFPs during the standstill state (Fig. 4h). During the running and standstill states, clear theta oscillations were detected in both LFPs as shown by the superimposed theta-filtered traces and their phase angles (Fig. 4f–i). Also, the single-unit traces recorded in the running state (Fig. 4g) exhibited significantly more bursting events than those in the standstill state (Fig. 4j).

LFP is generated from the merged signals by a large population of neurons, and a significant portion of neurons are phase-locked to a particular rhythm of LFPs, such as theta oscillation in the hippocampus[29,35]. To analyze the phase-locking of single-unit spikes in CA1, we separated spike waveforms by PCA clustering, after 500–3000 Hz bandpass filtering of raw signals recorded from this running mouse. Two different spike waveforms were detected, as shown in Fig. 4k, with the isolation distance of 112.6 and $L$-ratio of 0.03359 and 0.00819, demonstrating good unit separation. Also, each single-unit neuron indicated phase-locking at a distinct angle. As shown in Fig. 4l, m, Cluster 1 was phase-locked around 330° of theta phase with $p < 0.001$. Another unit (cluster 2) also showed phase-locking around 45° of theta phase with $p < 0.002$. Results from another mouse are presented in Supplementary Fig. 27.

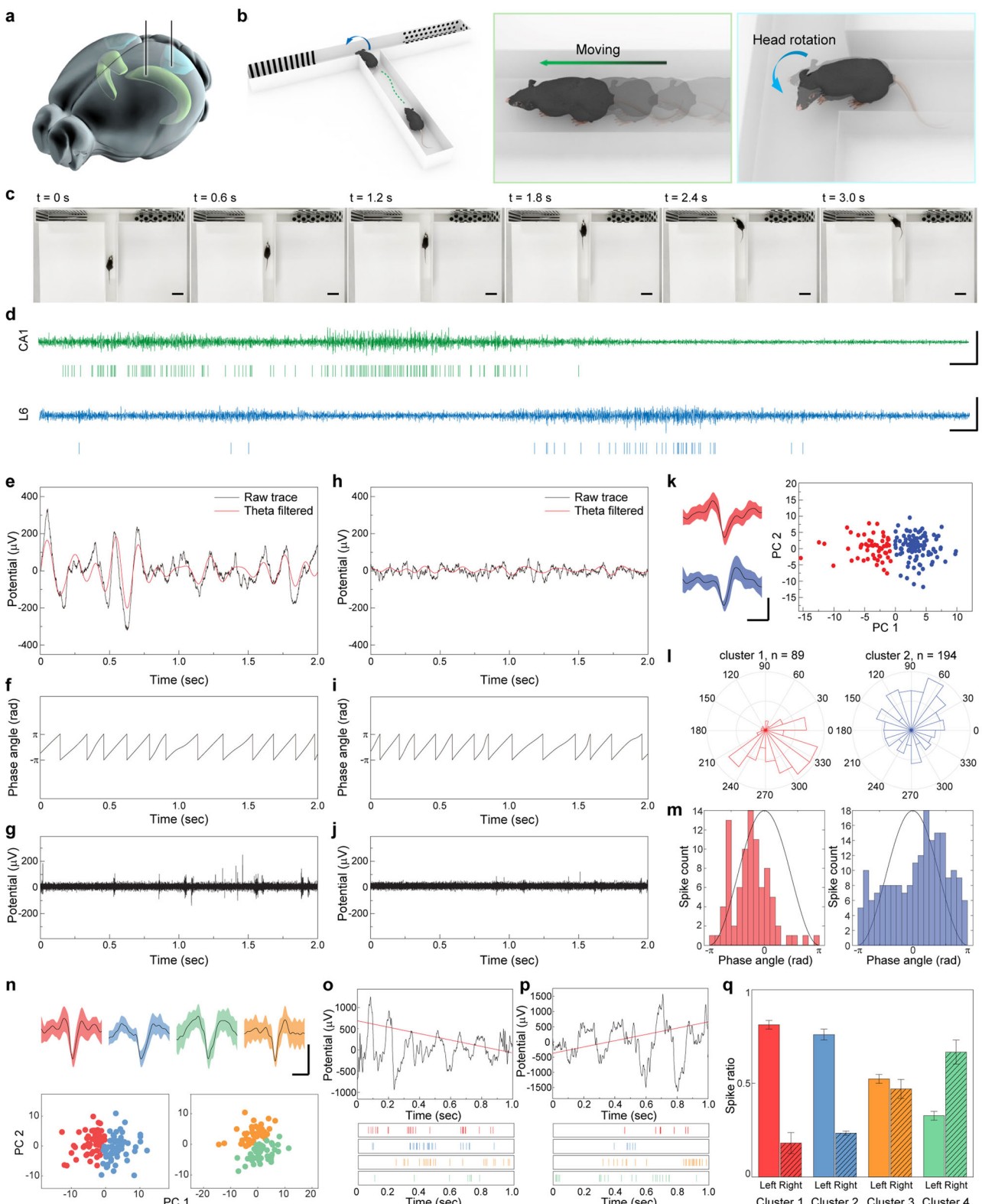

Throughout the visual system, the activity of neuronal signals is related to the motion of the object in view. However, to detect the motion trajectory and speed of an object, visual stimuli must occur within a context that includes information on the motion status of an observer. During the head-rotation motion, a stationary object is perceived as moving from the observer's point of view, and it can generate neuronal activities in the visual cortex[32]. We implanted two neural probes in the adjacent sites within the visual cortex L6 region, and four

different spike waveforms from two probes were clustered by PCA with the isolation distance of 268.6 and 102.8, and $L$-ratio of 0.0046, 0.0004 and 0.006, 0.003, respectively, which indicated a reliable separation between single units (Fig. 4n). We monitored the LFPs and single-unit spikes during the head-rotating behavior of a mouse with left- and right- directions, respectively. During its left-turn (counterclockwise) task, the recorded LFP trace showed the depolarization regime, and it appeared as the negative slope of linear fit (Fig. 4o). In

**Fig. 4 | Wireless neural recording of freely moving mouse. a** Schematic illustration of approximate locations of neural probes, primary visual cortex (blue) and hippocampus (green). Brain image was from Allen Mouse Brain Atlas. **b** Schematic illustration showing the behavior of mouse in T-maze, including the active movements and head-rotation motions. **c** Representative sequential snapshots of a freely moving mouse in T-maze for ten times of trials. Scale bars, 5 cm. **d** Representative single-unit traces and spike raster of the hippocampus CA1 and the primary visual cortex L6 regions during the T-maze test. Scale bars, 300 μV (vertical) and 100 ms (horizontal). Local field potential and superimposed theta waves (**e**), the corresponding theta angle (**f**), and single-unit trace (**g**) recorded from CA1 region during the active movement of mouse. Local field potential and superimposed theta waves (**h**), the corresponding theta angle (**i**), and single-unit trace (**j**) recorded from CA1 region during the standstill state of mouse. **k** PCA-clustered single-unit spikes recorded in CA1 region. Scale bars, 100 μV (vertical) and 1 ms (horizontal). Shaded regions represent 95% confidence intervals. **l** Circular distribution of theta angles for the firing events of each neuron recorded in CA1. **m** Distribution of single-unit spikes to theta phase angle. Black lines represent a theta cycle. **n** PCA-clustered single-unit spikes recorded in L6 region. Scale bars, 100 μV (vertical) and 1 ms (horizontal). Shaded regions represent 95% confidence intervals. Representative local field potential and corresponding spike raster of PCA-clustered neurons during left-turn task of mouse (**o**) and right-turn task of mouse (**p**). Red lines are linear fit of LFP signals. **q** Average spike ratio of PCA-clustered neurons between left-turn and right-turn tasks for five times of trials. Error bars represent the standard deviation ($n = 5$ independent left- and right-turn experiments of the mouse).

contrast, during its rotation to the right direction (clockwise), neuronal spikes were hyperpolarized and resulted in the increment of LFP (Fig. 4p). These dependencies of visual cortex signals to the head-rotating direction showed a similar tendency of previously reported spatial navigation signals in rodents[32,36]. For each PCA-clustered neuron, its spike ratio was compared between during the left-turn and right-turn behaviors of Fig. 4o, p. The cluster 1 and 2 neurons showed a biased spiking to left-turn event, and the spikes of cluster 4 neuron showed opposite bias to the cluster 1 and 2 neurons (Fig. 4q). The biased spiking of neurons can be regarded as a contextual dependency on the motion status of a subject for visual-motion processing in the visual cortex[32]. Results from another mouse are presented in Supplementary Fig. 28.

To determine whether there was any undesired damage due to the heat produced by the wireless neural recording system, the generation of heat was monitored using an infrared camera during the neural recording (Supplementary Movie 5). The temperature of the scalp with the circuit was maintained at an average of 37.1 °C with no significant fluctuation during the T-maze test. Given that the normal body temperature for mice is ranged from 36.5 to 38 °C, the circuit is considered to have negligible heat generation. Along with the thermal camera result, we did not observe any signs of abnormal or suffering behaviors in mice.

## Discussion

We developed a monolithic neural interface system that can monitor the single-unit activities of neurons with long-term stability using a high-resolution printing of liquid metal on the cranium. For this, neural probes of a low-modulus, biocompatible liquid metal was printed to the desired lengths with neuron-like diameters to reach the specific depth in the brain with minimal invasiveness. To enhance the signal quality of these neural probes, Pt nanoclusters were formed on the surface of neuron-interfacing liquid metals. Furthermore, direct printing of liquid-metal interconnections along the curvature of a cranial surface enabled the conformal integration of electronics to the cranium, and wireless neural recording using these printed cranial electronics was also demonstrated. Using this conformal high-resolution printing of liquid metal, multiple neural probes implanted in arbitrary brain regions were interconnected as the cranial electronics to form a monolithic neural interface system. The in-vivo studies demonstrated the long-term recording of LFPs and single-unit spikes in MO, HIP, and VIS regions of a live mouse with tracking of the same neurons for 33 weeks. The T-maze test using a freely moving mouse presented the behavior-induced activation of neurons in particular brain regions.

Both gallium and indium, the two elements in EGaIn (eutectic Gallium-Indium alloy), have no known physiologic drawback in the human body. To date, these elements have led to the development as diagnostic and therapeutic agents in medical fields, including metabolic bone disease[37], cancer[38–40], and infectious diseases[41]. Gallium is known to have minimal cytotoxicity and has negligible solubility in water (except as a salt). The United States Food and Drug Administration (FDA) has approved the use of gallium salts, such as gallium citrate, as medical imaging agents for imaging cancer and inflammation via intravenous injection[42]. Indium is also negligibly soluble in water, and the FDA has approved indium-based materials as clinical diagnostic agents for detecting endocrine tumors and infection[43]. Generally, metal salts exhibit enhanced biological interactions, so the biocompatibility of gallium and indium salts could be represented as a positive indicator for the biocompatibility of those metals. Recently, in-vivo toxicity studies showed that EGaIn is non-toxic based on histological and blood panel indications at high oral administration[44] or surgical implantation[45]. Therefore, liquid metals have allowed the development of a range of biomedical devices, including biosensors[46,47], blood vessel[48], and implantable devices[45,49]. Future considerations for the potential translation of our neural recording system and on-tissue printing method to biomedical applications include i) evaluation of systemic toxicity of liquid metals and devices for acute and chronic responses, and ii) comprehensive studies on biocompatibility including dose-related studies and long-term clinical observation (throughout the life cycle).

Recent studies on neural probes have focused on monitoring narrow intracerebral regions with densely packed electrodes. Although our approach may have utilized fewer channels than these state-of-the art probe designs, certain clinical and practical applications within the neuroscience and medical communities can require more variable placement of multiple neural probes across broad areas of the brain. For instance, in the case of neuropathic pain, different intracerebral areas exhibit distinct responses to medical treatment, and these areas are distributed throughout the cerebral cortex and deep brain regions. Additionally, for epilepsy patients, it is necessary to sparsely implant multiple probes into the brain to monitor and localize seizure activity. In such cases, it is crucial to acquire signals from multiple areas simultaneously in order to gain a comprehensive understanding of the underlying mechanisms of treatment, rather than relying on a monotonous probe with densely packed channels in the narrow region. Therefore our system can offer promising potential to significantly impact the field of biomedical engineering by enabling extensive monitoring of various brain areas and providing customizable electronics beyond the pre-fabricated and pre-packaged devices[3,50–53]. As an example, the simultaneous neural recording throughout the broad brain regions with coverage of motor cortices, hippocampal regions, and visual cortices at single-unit levels are demonstrated in Figs. 3 and 4. Supplementary Table 1 presents the comparison of our work to state-of-art methods. This printable neural interface system can offer new opportunities to biomedical and neuroscience research by its long-term, simultaneous analysis of multiple and broad regions of a brain with single-neuron resolution. Through the further development of multi-channeled soft neural probes and their tailored circuit printing, our approach can acquire neural signals with high spatial resolutions for broad areas.

The conventional neural interfaces, which have significant size and weight for unique anatomical structures with high curvature, can significantly interfere with natural behavior, impacting the accuracy of

behavioral studies. Our strategy includes high-resolution printing of conformal electronics and precise insertion of soft neural probes with monolithic materials, achieving flexibility in the choice of designs of the circuits ranging from planar lines to 3D structures within confined areas of the cranium in situ. Moreover, this direct printing on the cranial surface allows the formation of a thin, lightweight, and conformal neural interface system. This approach maximizes the structural integrity between the body and electronics to a level where suturing can be performed with minimal difference between pristine and post-formation states. When applied to small animals such as mice, which play a crucial role in preclinical studies of behavioral experiments, our strategy can achieve their highly free behavior by eliminating behavioral interference caused by the bulkiness and hindrance of the neural interface system. For large animals or humans (for recording or stimulating on multiple sites of cortex or deep structures), this high-resolution printing approach shows the potential to be performed at the stage of pre-planning, based on 3D models from pre-examined MRI.

However, short procedure time and high fabrication yield are preferred for human translation. Printing on the human brain, having a much larger surface area, requires improvements in printing speed compared to the small animal model cases. Also, it should provide high yield while forming larger circuits. We believe that such issues can be addressed through real-time surface profiling of the cranium by laser or depth camera[54], the selection of passivation layers with enhanced adhesion to liquid metals[55], and the modification of liquid metals' rheological properties[16] to allow high-yield, fast conformal printing.

We demonstrated the wireless neural recording using the wi-fi as well as NFC-based system by printing liquid metals into body-integrated cranial circuits. Through the NFC-based communication, we achieved a recording system conformal to the cranium by embedding the whole system into a scalp. On the other hand, the current state of wi-fi recording was still less integrative to the body, due to its larger size and bulky battery requirements. Printing batteries on the cranium using biofluids as electrolytes can be a follow-up study to achieve comprehensive integrity of wireless neural recording systems. Also, the ability to form soft neural interfaces and conformal circuits on the soft biological surface presented in this manuscript suggests various applications from neuroscience to bioelectronics and represents promising areas for future work.

## Methods

### Liquid metal printing

A glass capillary (Sutter Instrument) with an outer diameter of 1.0 mm and an inner diameter of 0.5 mm was pulled with a pipette puller (P-1000, Sutter Instrument) to prepare the nozzle which had an inner diameter of 5 μm. EGaIn (75.5% gallium, 24.5% indium alloy by weight; Changsha Santech Materials Co. Ltd.) was used as a liquid-metal ink. All printing steps of EGaIn were monitored using a microscope camera (QImaging Micropublisher 5.0 RTV, Teledyne Photometrics) to control the nozzle location from a substrate using a 6-axis stage (H-820 6-Axis Hexapod, Physik Instrumente) during the printing process. For the 5-μm-diameter nozzle, using a compressed dry air, the air pressure of 60 psi was applied to the liquid metal ink to be pulled from the syringe (i.e. an ink reservoir) to the tip of a glass capillary nozzle before starting printing. The line-width of the printed liquid metal was measured using SEM.

### Fabrication of soft neural probes

The fabrication steps of the soft liquid-metal neural probe are as follows: (1) a Si wafer (Dasom RMS, Republic of Korea) was spin-coated by a LOR 3 A lift-off resist (MicroChem) as a sacrificial layer. (2) A 1 μm-thick parylene-C layer was deposited as the bottom layer of a neural probe using a parylene coating system (NRPC-500, Nuritech Co. Ltd., Republic of Korea). (3) EGaIn was printed to 5-μm-wide lines on this

parylene-C layer. (4) Another 2.5-μm-thick parylene-C layer was deposited as the top layer of a neural probe. (5) The photoresist S1818 (MicroChem) was spin-coated and then photolithographically patterned for defining the probe shape with opening the tip area of electrodes. (6) The areas of parylene-C where the S1818 layer did not cover was etched away by $O_2$ plasma using a reactive ion etching (RIE) system (LAT Co. Ltd., Republic of Korea), and then the S1818 photoresist was dissolved out using acetone. (7) The S1818 photoresist was spun again and then photolithographically patterned for opening the electrode pads, and procedure (6) was repeated. (8) The resulting probes were lifted off using a remover PG solution (MicroChem) by dissolving the sacrificial layer (the LOR 3A lift-off resist) and by releasing the neural probes from the Si wafer. (9) The released neural probes were rinsed with deionized (DI) water.

### Platinum black electrodeposition

For preparing 50 mL of an electroplating solution, we mixed 50 mL of deionized water, 10 mg of lead acetate trihydrate (Sigma-Aldrich), and 0.5 g of platinum tetrachloride (Sigma-Aldrich) at room temperature. This electroplating solution was stirred under ultrasonic vibration for 20 min. The electroplating was performed by ion transfer between the cathode and anode in the Pt electroplating solution. A cathode (the soft neural probe that needs to be electroplated) and an anode (Ti/Pt electrode) were immersed in this electroplating solution, and each was connected to a source meter (Keithley 2400, Tektronix). The electroplating reaction occurred under an electrical current of 0.1 mA for 60 seconds.

### Printing of liquid metals on the cranium

For the printing of liquid metals on the cranium, a mouse was anesthetized and fixed on the adapter of the printing stage to minimize its motion or vibration during printing. And the nozzle (inner diameter: 5 μm) was adjusted to be ~5 μm above the targeted cranium surface, and a constant pneumatic pressure of 60 psi was applied during the printing. For Fig. 2a–c, 14-week-old male C3H mice (The Central Laboratory Animal Inc. Republic of Korea) were used. The mice were raised in a specific pathogen free (SPF) environment with an ambient temperature of 23 °C, a humidity of 50%, and a 12 h dark/light cycle.

### Electromechanical characterization

The stretching-induced fracture and self-healing test were performed using an uniaxially stretching stage and a step motor controller (SMC-100, Ecopia). The EGaIn pattern printed on a parylene-C film was attached to this stretching stage before its stretching, and both ends of this EGaIn pattern was connected to a source meter (Keithley 4200-SCS, Tektronix) for measuring its resistance. The movement of the stretching stage was manipulated by a 1-axis high-speed motion controller (PMC-1HS program, Autonics). The change in resistance during the self-healing event of this EGaIn sample was continuously measured with a sampling rate of 5 ms.

### Micro-computerized tomography (Micro-CT)

3D images of cranium with our printed circuit pattern were taken by high-resolution micro-CT (Skyscan 1173, Bruker) at 100 kVp and 100 μA source, and the images were reconstructed using CTAN software (Bruker).

### Impedance spectroscopy

The impedance measurements of the pristine EGaIn and PtB/EGaIn electrodes were conducted in a PBS solution (Sigma-Aldrich). For the in-vivo impedance measurement, neural probes with the PtB/EGaIn electrode were implanted into the brain of an awake, head-fixed mouse (C57BL/6N, male). All impedance measurements were performed over a frequency range of 0.01 to 100 kHz using a multichannel potentiostat (PMC-1000, AMETEK).

Impedance restoration test: The PtB/EGaIn side of our neural probe was soaked in PBS, and the upper part of a neural probe was folded to disconnect the electrical traces without the parylene-C fracture. After that, the neural probe was unfolded again and the impedance value was then measured using the potentiostat.

## Tensile tests
The elastic modulus of a liquid-metal based neural probe was measured by stress-strain characteristics during its tensile test using a TXA™ micro-precision texture analyzer (Yeonjin S-Tech, Republic of Korea). The strain rate was 10 μm/s during its stretching and releasing.

## Animal experiments
6-week-old male C57BL/6N and C3H mice (The Central Laboratory Animal Inc. Republic of Korea) were housed in SPF environment with 23 °C temperature and 50% humidity, with a 12 h light/12 h dark cycle. Sex was not considered in the study, and only male mice was used for controlling the variables of the experiment. For anesthetizing, 100 mg kg$^{-1}$ ketamine was used. All animal experiments were carried out in accordance with the recommendations in the Guide for the Care and Use of Laboratory Animals of the Yonsei University Institutional Animal Care and Use Committee. The protocols were approved by the Committee on the Ethics of Animal Experiments of Yonsei University (approval number IACUC-A-202011-1180-01).

## Neural recording
For the data analysis of LFPs and single-unit potentials, an electrophysiological recording system was used, which consisted of a RZ2 amplifier processor, PZ5 Neurodigitizer, MZ60 MEA interface (Tucker-Davis Technologies Inc, USA), and a computer with Synapse program. A sampling rate of 24,414 Hz and 60 Hz notch were used during recording. Mostly 0.1–300 Hz bandpass filter was used for recording LFPs, and a 300–6000 Hz or 500–3000 Hz bandpass filter was used for recording single-unit spikes. For in-vivo brain recording, soft neural probes were implanted to specific stereotaxic coordinates in the brain of an awake, head-fixed mouse (C57BL/6 N, male) using a 6-axis movement stage, and the 0–80 set screw was used as a reference. Each experiment was repeated with at least 3 different mice. The minimum animal numbers of each experiment ($n = 3$) were determined based on previous publications[3,56].

LFPs and single-unit spikes recorded from 12 soft neural probes: In Fig. 3, both LFPs and single-unit spikes were recorded according to the following stereotaxic coordinates: (1) Motor cortices (MO): 0.7 mm AP, 1.8 mm ML, 1.6 mm DV from bregma; 0.74 mm AP, 1.4 mm ML, 1.6 mm DV; 0.8 mm AP, −0.1 mm ML, 1.0 mm DV; 0.8 mm AP, −0.75 mm ML, 1.0 mm DV. (2) Hippocampal regions (HIP): −1.8 mm AP, 1.6 mm ML, 1.5 mm DV; −1.8 mm AP, 0.9 mm ML, 1.5 mm DV; −1.8 mm AP, −0.7 mm ML, 1.7 mm DV; −1.7 mm AP, −1.1 mm ML, 1.6 mm DV. (3) Visual cortices (VIS): −3.3 mm AP, 1.8 mm ML, 0.6 mm DV; −3.1 mm AP, 1.1 mm ML, 0.6 mm DV; −3.1 mm AP, −0.9 mm ML, 0.6 mm DV; −2.8 mm AP, −1.4 mm ML, −0.6 mm DV. AP, ML and DV represent anteroposterior, mediolateral, and dorsoventral, respectively. For the wi-fi wireless recording system, a wireless recording unit (JAGA16 and JAGA Penny, Jinga-hi, USA) was used.

## Tissue clearing
The tissue clearing of a mouse brain followed SHIELD protocol[27]. A mouse was transcardially perfused with an ice-cold phosphate-buffered saline (PBS) and then with the SHIELD perfusion solution. The dissected brain was incubated in the same perfusion solution at 4 °C for 48 h. The brain was then transferred to the SHIELD-OFF solution and incubated at 4 °C for 24 h. Then, the whole brain was split into two hemispheres. Following the SHIELD-OFF step, the brain was placed in the SHIELD-ON solution and incubated at 37 °C for 24 h. The SHIELD-fixed brain was cleared passively for a couple of weeks (10-14 d at

45 °C) in a sodium dodecyl sulate (SDS)-based clearing buffer (300 mM SDS, 10 mM sodium borate, 100 mM sodium sulfite, pH 9.0). The ice-cold PBS containing 4% (w/v) paraformaldehyde (PFA), 5 M NaCl, and GE38 supernatant (pH 7.2) was used as the SHIELD perfusion solution. The ice-cold PBS containing 5 M NaCl and GE38 supernatant (pH 7.2) was used as the SHIELD-OFF solution. 0.05 M sodium carbonate and 0.05 M bicarbonate were mixed and prewarmed to 37 °C for the use as the SHIELD-ON solution.

## Immunostaining
After clearing, the brain was rinsed with PBST (1 × PBS with 1% Triton X-100, 0.02% sodium azide) at RT for 3-4 times. The brain was then embedded in 4% agarose (Sigma-Aldrich), cut including the neural probe into blocks of 1 mm (length) ×2 mm (width) ×1 mm (height). The brain tissue was incubated with mouse anti-FOX3 (1:500, SIG-39860, BioLegend), rat anti-GFAP (1:300, 13-0300, Invitrogen), and rabbit anti-Iba1 (1:300, PA5-27436, Invitrogen) for 4 days at RT. After incubation, brain tissue was rinsed in fresh PBST to 3 times before its overnight incubation at RT. The brain tissue was subsequently incubated with donkey anti-mouse Alexa Fluor 488 (1:300, Invitrogen), donkey anti-rat Alexa Fluor 594 (1:300, Invitrogen), goat anti-rabbit Alexa Fluor 647 (1:300, Invitrogen) for 3 days at RT. Then this brain tissue was rinsed with PBST up to 3 times at RT. Finally, this brain tissue was incubated in a refractive index matching solution for 30 min, and then images were scanned using a fluorescent microscope. The minimum animal numbers of each experiment ($n = 6$) were determined based on previous publications[14,53].

## Fluorescence microscopy
Fluorescence images were acquired on a DM6 CFS fluorescent microscope (Leica, Germany) with a ×20 (0.5NA) objective. For the fluorescence microscopy of a horizontal section of brain tissues, sections of cleared brain tissue with immunostaining were visualized using an Axio Imager M2 (Carl Zeiss).

## LIVE/DEAD assay
Neuro2a cells, the neuroblast cell type, were used, and cell viability was evaluated by staining using the Live/Dead Imaging kit (LIVE/DEAD™ Cell Imaging Kit (488/570), Thermo Fisher Scientific) after 4 and 8 days of culture.

## Fluorescence intensity measurement
ImageJ software (version 1.53t) was used to quantify the fluorescence intensity profile. Fluorescence images were converted to 16-bit images for quantification of each pixel. To measure the intensity profile, linear selection was applied across the probe insertion area, and the linear profile of the selected region was acquired via the 'Plot Profile' function of this software.

## Haematoxylin and Eosin (H&E) staining
Brain tissues were immersed in 10% formalin and then dehydrated with ethanol gradient. Subsequently, the tissue was embedded in paraffin, sectioned and stained with Haematoxylin (Sigma-Aldrich) and Eosin (Sigma-Aldrich). The staining was visualized by Axio Imager M2 (Carl Zeiss).

## T-maze test
The apparatus was a standard T-maze with one start arm (35 cm × 5 cm × 15 cm) and two finish arms (30 cm ×5 cm ×15 cm each). This maze was made of an acrylic plate with a white floor and sides. Two different patterns (line and circle) were attached to the walls of two goal arms. A mouse of 12 weeks post-injection was used. For T-maze behavioral tests, neuronal activities were recorded during T-maze behavioral experiments according to the following stereotaxic coordinates: (1) Hippocampal region (CA1): anteroposterior, −1.94 mm; mediolateral,

1.70 mm; dorsoventral, 1.35 mm. (2) Visual cortex (L6): anteroposterior, −3.5 mm; mediolateral, 2.4 mm; dorsoventral, 1.5 mm. Each experiment was repeated with at least 3 different mice. With each mouse, the test was repeated for ten times, and the representative data have been presented.

## Data analysis

Spike detection and PCA clustering: all analyses of neural recording data were done using MATLAB 2021a with four open-source toolboxes (Statistics and Machine Learning Toolbox, Signal Processing Toolbox, Bioinformatics Toolbox, Circular Statistics Toolbox by Philipp Berens). The threshold for spike detections was set as −5 times of the standard deviation of the filtered (300–6000 Hz or 500–3000 Hz bandpass) time series, and PCA was used for the reduction of the dimension by the *k-means* function.

Calculation of *L*-ratio and isolation distance: Both measurements were calculated based on the Mahalanobis distance. The *L*-ratio was calculated as follows:

$$L_{ratio} = \frac{L(C)}{N(C)} = \frac{\sum_{i \notin C} 1 - CDF_{\chi^2}(MD_{i,c}^2)}{N_c} \tag{1}$$

where $N_c$ is the number of spikes in cluster $C$, $i \notin C$ is the set of spikes which are not members of the cluster $C$, $CDF_{\chi^2}$ presents the cumulative distribution function of the $\chi^2$ distribution in an eight-dimensional feature space, and $MD_{i,c}^2$ is the Mahalanobis distance of spike $i$ from $C$. An $L$ ratio of less than 0.05 indicates good separation and isolation of the cluster. The isolation distance of a cluster $C$ that contains $N_c$ spikes were defined as the $MD^2$ value of the $N_c^{th}$ closet noise spike.

Phase-locking analysis: 4–8 Hz bandpass was applied to filter the theta oscillation. The distribution of theta phases at spiking events was determined for all clusters with at least 25 single-unit spikes. Rayleigh criterion was used to test the nonuniformity of theta phase distributions using the open-source Circular Statistics Toolbox for MATLAB.

Fluorescence image-based quantification: the quantification of fluorescence intensity of confocal micrographs was conducted with the ImageJ software.

## Statistical analysis

All data were presented as mean ± standard deviation (S.D.). Statistic calculations of *p*-value were performed to analyze the phase locking, using an open-source function (channelobject.uniformTest()) of paired *t*-test in MATLAB.

## Statistics and reproducibility

The experiments in Fig. 1e, f were repeated 30 or more times with similar results, including the neural probes for multi-channel recordings presented in Figs. 3 and 4. The experiment in Fig. 2b, c was repeated from 3 times with similar results. The experiment in Fig. 2c, d was repeated 5 times with similar results. The experiment in Fig. 3b was repeated 3 times with similar results.

## Reporting summary

Further information on research design is available in the Nature Portfolio Reporting Summary linked to this article.

## Data availability

The main data supporting the results in this study are available within the paper and its Supplementary Information. Any additional requests for information can be directed to, and will be fulfilled by, the corresponding authors. Source data are provided with this paper. Source data are available at Figshare (https://doi.org/10.6084/m9.figshare.24112461)[57]. Source data are provided with this paper for reproducing all Figures in the manuscript and Supplementary Information. We used the Allen Mouse Brain Atlas to identify and illustrate the brain's stereotaxic coordination (Allen Mouse Brain Atlas. Available from https://mouse.brain-map.org). Source data are provided with this paper.

## Code availability

The custom code used for this study is available at Code Ocean (https://doi.org/10.24433/CO.1696883.v1)[58].

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

## Acknowledgements

This work was supported by the Ministry of Science & ICT (MSIT), the Ministry of Trade, Industry and Energy (MOTIE), the Ministry of Health & Welfare, and the Ministry of Food and Drug Safety of Korea through the National Research Foundation (2023R1A2C2006257) (J.-U.P.), ERC Program (2022R1A5A6000846) (J.-U.P.), and the Korea Medical Device Development Fund grant (RMS 2022-11-1209 / KMDF RS-2022-00141392) (J.-U.P.). This work was also supported by Institute for Basic Science (IBS-R026-D1) (J.-U.P.). We thanks to Prof. Chang Ho Sohn and Dr. Mi Jung Kim at Yonsei University for their valuable instruction in the tissue clearing process.

## Author contributions

Y.-G.P., Y.W.K., and E.K. carried out the experiment, analyzed the data, and wrote the manuscript. C.S.K and Y.-M.H. were involved in all animal experiments and the related analysis. D.H.L., S.K., J.-Y.K., and S.L. were involved in device fabrications, and J.M. performed the MATLAB coding. J.-U.P., J.W.C., J.C., H.H.J., and J.-H.L. oversaw all of the research phases and revised the manuscript. All authors discussed and commented on the manuscript.

## Competing interests

The authors declare no competing interests.
