## [Peer Review File · Nature Communications]

REVIEWER COMMENTS

Reviewer Reviewer #1

The authors have made some revisions to manuscript 22-2411. I appreciate their efforts trying to address mine and other reviewers' comments. However, after very carefully reading the revised manuscript, I have to stand by my previous assessment, I did not find sufficient evidence that can justify the authors' claims. Besides, some of the new data were obviously flawed.

Overall, it is still fair to state that the only novelty in this paper is a method to print liquid metal and its insulation on the skull surface, which is interesting but hardly useful the the described applications in this manuscript. In the revised manuscript, no convincing data or argument were shown to sufficiently support the claimed advantages, while there are major drawbacks.

Simple conventional solutions can easily do all the demonstrated experiments, and most likely easier, of lower cost and much higher throughput. For example, pre-fabricated flex-pcb connected sensors and electrodes. These standard solutions can easily make ad hoc customizations very easily use fast prototyping fabrication strategies, such as laser cutting. Direct writing interconnections, in contrast, has the intrinsic drawback of doing the engineering and fabrication during surgery, which unnecessarily makes the procedure time much longer. I have thought very hard but failed to think of any realistic scenario where this method is clearly advantageous over conventional methods.

The added experiment to compare Au and EGaIn traces, Fig S12, is absolutely misleading. The authors intended to indicate that their printed traces are better in "transmitting the electrical signals". There is no explanation why they designed the experiment this way and there is no modeling or calculation to interpret the data. Yet they reached a conclusion that "signal loss can occur during transmission across the Au interconnection. In contrast, the EGaIn electrode did not exhibit a significant delay in the transient response time, and as a result, the amplitude of the acquired signal was about 1.5 times higher than that of the Au".

This 'Voltage Transient' measurement is conventionally used to characterize the electrochemical performance of electrode contact material. (PtB in this case) Most likely, the measured results have nothing to do with either Au or liquid metal traces.

In Fig 1d, the authors made a schematic drawing to compare the sizes of neurons and their recording electrode. The soma is labeled '20-50 μm ' and the axon is labeled '5-20 μm '. There is no reference where

these numbers come from, but the authors must have made a mistake here. In reality, their electrodes are about 2-5 times bigger than the soma size, which is also clearly shown in their own histology images, whereas the axons are much thinner.

The demonstration experiment in Supplementary Fig. 5 was done in PBS, which has very different from brain tissues. This should be tested in vivo, or at least in materials similar to tissues, such as hydrogel. Displacement, success yield, etc, should be quantified and presented.

reviewer 4

The authors have reported a soft neural interface incorporating printable electronics on the cranial surface. This is an interesting and unique approach within the broader set of recent technological advances in the soft electrode design space.

The authors have made a dedicated effort to address critiques in the prior review. However, a few major concerns remain:

(1) A primary concern is that the number of animals enrolled in the in vivo assessments is insufficient to have confidence in the results reported. As an example, the SNR comparison reported in line 322-23 seems to be from 1 animal in each condition. It is inappropriate to report an improvement in SNR on the basis of a single animal subject. Generally, the in vivo recording results are examples from single animals or a small number of animals.

(2) The authors added a mechanical test to assess the robustness of their device in response to a prior critique. However, the test chosen--compressive force of 5 N, which is "20 times the mass of an average adult mouse"--has no basis as a physiologically relevant mechanical test for a brain implant (line 270-72).

Additional concerns include:

(3) Lines 377-78 indicate that increasing signal quality may be related to "the gradual alleviation of astrocytes and microglia near the probe as the tissue recovered from the acute damage," but gliosis typically does not alleviate over time. It is recommended that the authors revise this statement.

(4) Lines 753-4 indicate that, "Statistic calculations of p value were performed using an open-source function(channelobject.uniformTest()) of MATLAB," but no details on what test this function performs, or where/how it was applied to the data, are given.

(5) Generally, paragraph structure needs to be revised for readability (several very long paragraphs throughout the manuscript).

Reviewer #5

The Authors have made some efforts to address my concerns, but there are still certain aspects of their responses that remain open to debate. These aspects raise important questions about the overall level of advancement presented in the manuscript. Furthermore, the in-surgery direct-writing of interconnections is perceived as impractical and unreliable for real-world clinical applications.

(1) Concerns about the potential toxicity of the liquid metal. The authors' argument that the FDA's approval of gallium citrate (a salt form) implies the safety of gallium metal doesn't hold. It's essential to recognize that metals and their corresponding salts can have entirely different biological effects, as exemplified by the differences between sodium and sodium chloride. I would recommend that the authors either conduct a systematic evaluation of the toxicity of both gallium and indium metals or provide examples of FDA-approved implants that utilize soft-packaged gallium and indium metals to address these concerns.

(2) Concerns about the limited recording capability of the devices. Each device has a relatively small number of channels (12 channels), which restricts their ability to record from only a small number of neurons in each animal. Notably, in 2017, Fu et al. have demonstrated stable long-term recording using 128-channel flexible mesh device in each animal (PNAS, 114, E10046-E10055, 2017). More recently, used flexible probes and achieved month-long stable recordings from approximately 1000 neurons in each animal (Nature Biomedical Engineering, doi.org/10.1038/s41551-022-00941-y). These examples indicate the need for the authors to consider how to enhance the recording capacity of their devices in order to compete with current advancements in the field.

(3) Concerns about the potential impacts of the current in-surgery direct-writing approach. This method comes with several drawbacks, including reliability issues and an increase in surgical time. It's important to note that all fabrication methods have yield issues, but prefabricated devices have the advantage of allowing for the elimination of defective products before implant surgery, reducing risks to the patient. In contrast, the use of direct written interconnection devices significantly raises the risk of failed implanted devices, which is a considerable concern. Additionally, the increased surgical time associated with this approach can lead to increased intraoperative risks. These aspects warrant careful consideration and further evaluation of the practical feasibility and value of the in-surgery direct-writing method.

[Response to Reviewer #1]

We thank the reviewer for a thoughtful review of and comments about our manuscript, and we welcome the opportunity to address and clarify the issues raised in the referee report. Our responses to the points raised in the report are as follows:

Comment 1: “Overall, it is still fair to state that the only novelty in this paper is a method to print liquid metal and its insulation on the skull surface, which is interesting but hardly useful the the described applications in this manuscript. In the revised manuscript, no convincing data or argument were shown to sufficiently support the claimed advantages, while there are major drawbacks. Simple conventional solutions can easily do all the demonstrated experiments, and most likely easier, of lower cost and much higher throughput. For example, pre-fabricated flex-pcb connected sensors and electrodes. These standard solutions can easily make ad hoc customizations very easily use fast prototyping fabrication strategies, such as laser cutting. Direct writing interconnections, in contrast, has the intrinsic drawback of doing the engineering and fabrication during surgery, which unnecessarily makes the procedure time much longer. I have thought very hard but failed to think of any realistic scenario where this method is clearly advantageous over conventional methods.”

Response to Comment 1:

Our method is not intended to compete with the conventional solutions using pre-fabricated devices, but we expect that our printing approach can be applied to areas where it is difficult to use the pre-fabricated devices. As the reviewer mentioned, the pre-fabricated devices can offer easy accessibility with their lower cost by mass production and higher throughput capabilities. However, these pre-fabricated circuits have bulky connector designs and relatively heavy subsidiary circuits. Some unusual cases can have limited physical space in the cranium and unique anatomical structures with high curvature. The conventional neural interfaces, which have significant size and weight for these unique anatomical structures, can significantly interfere with natural behavior, impacting the accuracy of behavioral studies.

Our strategy includes high-resolution printing of conformal electronics and precise insertion of soft neural probes with monolithic materials, achieving flexibility in the choice of designs of the circuits ranging from planar lines to 3D structures within confined areas of the cranium in situ. Moreover, this direct printing on the cranial surface allows the formation of a thin, lightweight, and conformal neural interface system. This approach maximizes the structural integrity between the body and electronics to a level where suturing can be performed with minimal difference between pristine and post-formation states. When applied to small animals such as mice, which play a crucial role in preclinical studies of behavioral experiments, our strategy can achieve their highly free behavior by eliminating behavioral interference caused by the bulkiness and hindrance of the neural interface system.

For large animals or humans (for recording or stimulating on multiple sites of cortex or deep structures), this high-resolution printing approach can be performed at the stage of pre-planning, based on 3D models from pre-examined MRI. Although we have studied over the cranial vault of mice in this manuscript, our approach offers significant potential for printing the liquid metal-based circuitry on the cortex level fitted along the groove of sulci-gyri patterns. This method can be applied as individually tailored, personalized neural interface system.

Revised Manuscript (page 25-26):

The conventional neural interfaces, which have significant size and weight for unique anatomical structures with high curvature, can significantly interfere with natural behavior, impacting the accuracy of behavioral studies. Our strategy includes high-resolution printing of conformal electronics and precise insertion of soft neural probes with monolithic materials, achieving flexibility in the choice of designs of the circuits ranging from planar lines to 3D structures within confined areas of the cranium in situ. Moreover, this direct printing on the cranial surface allows the formation of a thin, lightweight, and conformal neural interface system. This approach maximizes the structural integrity between the body and electronics to a level where suturing can be performed with minimal difference between pristine and post-formation states. When applied to small animals such as mice, which play a crucial role in preclinical studies of behavioral experiments, our strategy can achieve their highly free behavior by eliminating behavioral interference caused by the bulkiness and hindrance of the neural interface system. For large animals or humans (for recording or stimulating on multiple sites of cortex or deep structures), this high-resolution printing approach shows the potential to be performed at the stage of pre-planning, based on 3D models from pre-examined MRI.

Comment 2: “The added experiment to compare Au and EGaIn traces, Fig S12, is absolutely misleading. The authors intended to indicate that their printed traces are better in “transmitting the electrical signals”. There is no explanation why they designed the experiment this way and there is no modeling or calculation to interpret the data. Yet they reached a conclusion that “signal loss can occur during transmission across the Au interconnection. In contrast, the EGaIn electrode did not exhibit a significant delay in the transient response time, and as a result, the amplitude of the acquired signal was about 1.5 times higher than that of the Au”. This ‘Voltage Transient’ measurement is conventionally used to characterize the electrochemical performance of electrode contact material. (PtB in this case) Most likely, the measured results have nothing to do with either Au or liquid metal traces.”

Response to Comment 2:

In the in-vivo recording, it is challenging to quantitatively compare the signal quality from the amplitudes and waveforms of single-unit signals propagated through probes because different neurons generate different signals. Therefore, we artificially generated controlled pulse signals mimicking single-unit spikes of neurons and compared the signal quality quantitatively by transmitting them to the signal recording unit through both existing materials (Au interconnect) and our printed EGaIn interconnect. Additionally, we placed two probe-interconnect pairs in close proximity to investigate potential noise coupling and other factors when receiving signals from multiple channels. To facilitate understanding of our designed in-vitro setup, we have indicated in Supplementary Fig. 10 how each component corresponds to aspects in the in-vivo setting and have modified the explanation of this experiment for easier comprehension.

Supplementary Fig. 10 (revised). Signal quality test with in-vitro setup mimicking the signal recording process from the brain to analyzer. **a**, Schematic illustration of the experimental setup for the signal quality test. **b**, Plots of acquired potential waveforms through a heterogeneous (soft neural probe – evaporated Au) pair (left) and through a monolithic (soft neural probe – printed EGaIn) pair (right).

Revised Manuscript (page 13-14):

We compared the signal recording quality of our monolithic connection (EGaIn neural probe – EGaIn interconnect) to heterogeneous connection (EGaIn neural probe – Au interconnect) through an in-vitro

setup that mimicked the signal recording process from the brain to analyzer via a neural probe and an interconnect (Supplementary Fig. 10a). Here, our soft neural probes were connected to evaporated Au or printed EGaIn interconnects (line-width: 5 μm , pitch: 100 μm), respectively. These neural probe-interconnect pairs were then assessed for signal quality by transmitting the controlled electrical signals (i.e. biphasic current pulse of 0.5 mA amplitude, 0.2 ms pulse width, and 0.02 ms interphase interval) to each pair connected with a potentiostat. (Here, biphasic current pulses were transmitted to generate the conditions similar to those for signals to pass through brain tissue to our probes and interconnections.) As shown in Supplementary Fig. 10b, the monolithic pair acquired 1.5 times higher potentials compared to the heterogeneous pair, possibly due to the minimal contact resistance between the monolithic pair of EGaIn neural probe and EGaIn interconnect. Also, our monolithic pair recorded potentials with less distortion of the injected pulse waveform.

To test the signal quality, a pair of Au and EGaIn electrode lines (line width: 5 μm , pitch: 100 μm) was formed on a polyimide film using photolithography and direct printing, respectively. Subsequently, these electrodes were subjected to the evaluation of signal quality by transmitting the electrical signals (i.e. biphasic current pulses) from a signal generator (3390, Keithley) to each electrode connected with a potentiostat. (Here, biphasic current pulses were transmitted from a signal generator to make the conditions similar to those for signals to pass through the brain tissue to our probes and interconnections.) Supplementary Fig. 11 shows a schematic illustration of this experimental setup. For the Au electrode, a delay in transient response time was observed during the cathodic pulse. This delayed transient response time indicates reduced sensitivity to instantaneous signal changes, which means signal loss can occur during transmission across the Au interconnection. In contrast, the EGaIn electrode did not exhibit a significant delay in the transient response time, and as a result, the amplitude of the acquired signal was about 1.5 times higher than that of the Au (Supplementary Fig. 12).

Revised Supplementary Information (page 13):

Supplementary Fig. 10. Signal quality test with in-vitro setup mimicking the signal recording process from the brain to analyzer. **a**, Schematic illustration of the experimental setup for the signal quality test. **b**, Plots of acquired potential waveforms through a heterogeneous (soft neural probe – evaporated Au) pair (left) and through a monolithic (soft neural probe – printed EGaIn) pair (right). **A** schematic

illustration of the experimental setup for signal quality test. Biphasic current pulses were transmitted from a signal generator through PBS to make the condition in which signals from the brain were transmitted through cranial interconnections. Then, the terminal of electrodes was immersed in PBS, and the signal waveforms were recorded using a potentiostat.

Revised Supplementary Information (page 14):

Supplementary Fig. 11. A schematic illustration of the experimental setup for signal quality test. Biphasic current pulses were transmitted from a signal generator through PBS to make the condition in which signals from the brain were transmitted through cranial interconnections. Then, the terminal of electrodes was immersed in PBS, and the signal waveforms were recorded using a potentiostat.

Revised Supplementary Information (page 15):

#1 Conventional interconnections – evaporated Au

#2 EGaIn interconnections – printed EGaIn

~~Supplementary Fig. 12. Signal quality tests using Au interconnections (evaporated) and EGaIn interconnections (directly printed) for our neural probes.~~

Comment 3: “In Fig 1d, the authors made a schematic drawing to compare the sizes of neurons and their recording electrode. The soma is labeled ‘20-50 μm’ and the axon is labeled ‘5-20 μm’. There is no reference where these numbers come from, but the authors must have made a mistake here. In reality, their electrodes are about 2-5 times bigger than the soma size, which is also clearly shown in their own histology images, whereas the axons are much thinner.”

Response to Comment 3:

As the reviewer suggested, we have included references for the size of each neuron part. Also, the depiction of neuron size in Fig. 1d has been modified, considering potential differences in size across species and individuals. In histology images, boundaries were indicated for voids caused by the insertion of the glass capillary, which did not align with the actual size of the neural probe. For the histology images in Fig. 3l-m and Supplementary Fig. 22, boundaries have been redefined to accurately represent the actual size of our neural probe.

Revised manuscript (page 7):

The soft neural probes had an electrode of 5-μm in line-width, and it is comparable to the widths of axons that carry action potentials³.

Revised manuscript (page 38):

3. Yang, X. *et al.* Bioinspired neuron-like electronics. *Nat. Mater.* **18**, 510–517 (2019).

Revised manuscript (page 44-45):

Figure 1 | d, Schematic illustration of a structural similarity between the soft neural probe and neuron.

Revised manuscript (page 48-49):

Figure 3 | l-m, 3D-reconstructed confocal micrograph (**l**) and fluorescence micrograph (**m**) showing implanted neural probes in hippocampal region. Yellow lines indicate the boundary of the implanted neural probe. Scale bars, 100 μm.

Revised Supplementary Information (page 27):

Supplementary Fig. 22. Color-blind safe images of 3D-reconstructed confocal micrograph and fluorescence micrograph showing the probe implanted in the hippocampal region. Scale bars, 100 μm .

Comment 4: “The demonstration experiment in Supplementary Fig. 5 was done in PBS, which has very different from brain tissues. This should be tested *in vivo*, or at least in materials similar to tissues, such as hydrogel. Displacement, success yield, etc, should be quantified and presented.”

Response to Comment 4:

We agree that the demonstration in PBS may differ from actual brain conditions. Therefore, we fabricated a brain phantom using a 0.6% agarose gel with mechanical properties similar to brain tissue. Results from these tests confirmed that our neural probe maintained a consistent position with displacements of less than 5 μm . We have modified Supplementary Fig. 14 with this new result.

[**Supplementary Fig. 14 (revised).** Sequential snapshots of our neural probes released from a glass capillary with matching the retraction velocity and volumetric flow rate (top). This capillary moved upward, while the end of the probe remained stationary (red dashed line). As a result, the neural probe maintained a consistent position with a displacement of less than 5 μm . In this experiment, 0.6% agarose gel was used as a brain phantom. Scale bars, 100 μm .]

Revised Supplementary Information (page 19-20):

Supplementary Fig. 14. Sequential snapshots of our neural probes released from a glass capillary with matching the retraction velocity and volumetric flow rate (top). This capillary moved upward, while the end of the probe remained stationary (red dashed line). As a result, the neural probe maintained a consistent position with a displacement of less than 5 μm . In this experiment, 0.6% agarose gel was used as a brain phantom. Scale bars, 100 μm . Schematic illustrations describing the procedures to implant a soft neural probe with a capillary and to connect the probe using liquid-metal interconnections (bottom). ~~Sequential snapshots of neural probes released from the glass capillary with matching the retraction velocity and volumetric flow rate. The glass capillary moved upward (blue dashed arrow), while the end of the probe remained stationary (red dashed arrow). Scale bars, 100 μm (top). Schematic illustrations describing the procedures to implant a soft neural probe with a capillary and to connect the probe using liquid-metal interconnections (bottom).~~

[Response to Reviewer #4]

We thank the reviewer for a thoughtful review of and comments about our manuscript, and we welcome the opportunity to address and clarify the issues raised in the referee report. Our responses to the points raised in the report are as follows:

Comment 1: “A primary concern is that the number of animals enrolled in the in vivo assessments is insufficient to have confidence in the results reported. As an example, the SNR comparison reported in line 322-23 seems to be from 1 animal in each condition. It is inappropriate to report an improvement in SNR on the basis of a single animal subject. Generally, the in vivo recording results are examples from single animals or a small number of animals.”

Response to Comment 1:

We calculated the signal-to-noise ratio for single-unit potentials obtained from three different mice, and revised Supplementary Fig. 11. In case of representative neural recording data shown in the main text, we provided the data from different subjects in Supplementary Information, including Supplementary Fig. 5, 13, 26, 27, and 28.

Revised manuscript (page 14-15):

After obtaining signals from each probe, we calculated the signal-to-noise ratio (SNR), the ratio of root-mean-squares (rms) of signals to that of noises. For the conventional system, the SNR was calculated to be 3.483 **in average**. In comparison, our system's SNR was 5.977 **in average**, indicating approximately 1.5 times improvement in signal quality compared to the conventional system.

Revised Supplementary Information (page 16):

Supplementary Fig. 11. Representative recorded single-unit traces and signal-to-noise ratios from (i) conventional Nichrome probes with PCB connectors (top) and (ii) our soft neural probe with EGaln interconnections (bottom). Both systems were each interfaced with a wi-fi module for data transmission. Signals were recorded from mice 4 weeks after implantation. Each error bar in signal-to-noise ratio plot represents a standard deviation of measurements from three different mice.

Comment 2: “The authors added a mechanical test to assess the robustness of their device in response to a prior critique. However, the test chosen--compressive force of 5 N, which is “20 times the mass of an average adult mouse”--has no basis as a physiologically relevant mechanical test for a brain implant (line 270-72).”

Response to Comment 2:

We aimed to demonstrate the device robustness by applying sufficient force (5 N) without causing any issues with device failure when applied in the subject. However, to make the compressive force physiologically relevant, we considered potential impact ranges during the normal behavior or group living of subjects, and we rearranged the force presentation to 1 N.

Revised manuscript (page 12-13):

To test the mechanical stability, an NFC-based wireless circuit was formed on a curved structure using our direct printing method (Supplementary Fig. 8a), and evaluated the extent of circuit deformation under a compressive force of 1 N was applied to the sample (Supplementary Fig. 8b). This is approximately 3 times the weight of an adult mouse (~ 0.3 N), in consideration of the impact that may occur during the normal behavior or group living of subjects. ~~The entire fabrication procedure of this sample followed the method of printing and encapsulating the cranial interconnections, as shown in~~

Supplementary Fig. 8. The fabricated sample was subjected to a compressive force of 5 N (through a compression tester machine (Mark 10)), which was equivalent to 20 times the mass of an average adult mouse. During the force application process, no observable deformation in the overall shape of this sample was detected.

Revised Supplementary Information (page 10-11):

Supplementary Fig. 8. Mechanical compression test of printed EGaIn circuit. **a**, Schematic illustration showing the entire fabrication procedure of a sample for mechanical compression test. **b**, Photographs of the printed wireless circuitry captured before and during compression at 1 N, 0 N, 1 N, and 5 N. Scale bars, 5 mm. **c**, Stereomicrographs before (left) and after (right) this mechanical compression. Scale bars, 1 mm. ~~A schematic illustration showing the entire fabrication procedure of the sample for compression test. **a**, Photographs of the wireless circuit captured during compression at 0 N, 1 N, and 5 N. Scale bars, 5 mm. **b**, Stereomicrographs of the wireless circuit before (left) and after compression (right). Scale bars, 1 mm.~~

Comment 3: "Lines 377-78 indicate that increasing signal quality may be related to "the gradual alleviation of astrocytes and microglia near the probe as the tissue recovered from the acute damage," but gliosis typically does not alleviate over time. It is recommended that the authors revise this statement."

Response to Comment 3:

Thank you for the comment. We revised the manuscript by removing the statement about the alleviation of gliosis.

Revised manuscript (page 18):

And then this amplitude gradually increased to reach ~143 μV at 15 weeks post-implantation, possibly ~~due to the gradual alleviation of astrocytes and microglia near the probe~~ as the tissue recovered from the acute damage³.

Comment 4: "Lines 753-4 indicate that, "Statistic calculations of p value were performed using an open-source function(channelobject.uniformTest()) of MATLAB," but no details on what test this function performs, or where/how it was applied to the data, are given."

Response to Comment 4:

The function performed paired t-test for statistical analysis, and we used this to analyze phase locking. We added the above information in the revised manuscript.

Revised manuscript (page 36):

All data were presented as mean \pm standard deviation (S.D.). Statistic calculations of p-value were performed ~~to analyze the phase locking~~, using an open-source function (channelobject.uniformTest()) of ~~paired t-test~~ in MATLAB.

Comment 5: "Generally, paragraph structure needs to be revised for readability (several very long paragraphs throughout the manuscript)."

Response to Comment 5:

We have divided long paragraphs in the manuscript to enhance the readability of the text.

Revised manuscript (page 7):

This fine area of our liquid-metal electrode (the overall surface area of this exposed cylinder: 58.8 μm^2) results in high spatial resolution for single-neuron-scale recording.

To further enhance the signal quality of this small opening of an electrode, we deposited platinum nanoclusters, denoted as platinum black (PtB)¹⁴, only at the open area of this liquid-metal electrode by electrodeposition (Fig. 1f).

Revised manuscript (page 9):

Also, some latest studies of these liquid metals also revealed their negligible toxicity and biocompatibility²⁰.

Figure 2a and Supplementary Video 1 show a conformal 3D printing of EGaIn directly on the cranium of a living mouse.

Revised manuscript (page 10):

The printing velocity was 80 $\mu\text{m/s}$ and the total printing duration was 22 minutes, which was within the anaesthesia time of a mouse.

Figure 2d and 2e show a SEM and an optical micrograph of the near-field communication (NFC) chip and chip antenna interconnected by printing free-standing 3D structures of EGaIn directly on a cranial surface of a mouse.

Revised manuscript (page 12):

Additional wireless LFP signals from the other two mice are plotted in Supplementary Fig. 5. After the anaesthesia wore off, the mouse showed its healthy movement.

To ensure the stability of the cranial circuit in mice, the circuit was visualized using micro-computerized tomography (micro-CT) 6 weeks after the circuit formation.

Revised manuscript (page 13):

The relative change in electrical resistance of this sample was negligible (Supplementary Fig. 10).

We compared the signal recording quality of our monolithic connection (EGaIn neural probe – EGaIn interconnect) to heterogeneous connection (EGaIn neural probe – Au interconnect) by in-vitro setup mimicking the signal recording process from the brain to analyzer via a neural probe and an interconnect (Supplementary Fig. 11a).

Revised manuscript (page 15):

In comparison, our system's SNR was 5.977 in average, indicating approximately 1.5 times improvement in signal quality compared to the conventional system.

The adaptability of direct printing effectively addresses individual variances in the shape and size of the brain and cranium, allowing customizable circuit configurations for specific needs.

Revised manuscript (page 17):

Figure 3c illustrates the top-view of the printed neural interface system in Fig. 3b, and indicates the relative position of neural probes (as black dots) and their cranial interconnections.

The soft neural probes exhibited low impedance of 412.0 $\text{k}\Omega$ at 1 kHz on average, with a standard deviation of 13.3 $\text{k}\Omega$, presenting the good uniformity in impedance of multiple neural probes (Fig. 3d).

[Response to Reviewer #5]

We thank the reviewer for a thoughtful review of and comments about our manuscript, and we welcome the opportunity to address and clarify the issues raised in the referee report. Our responses to the points raised in the report are as follows:

Comment 1: “Concerns about the potential toxicity of the liquid metal. The authors' argument that the FDA's approval of gallium citrate (a salt form) implies the safety of gallium metal doesn't hold. It's essential to recognize that metals and their corresponding salts can have entirely different biological effects, as exemplified by the differences between sodium and sodium chloride. I would recommend that the authors either conduct a systematic evaluation of the toxicity of both gallium and indium metals or provide examples of FDA-approved implants that utilize soft-packaged gallium and indium metals to address these concerns.”

Response to Comment 1:

We agree with the concern that liquid metals have not been frequently used in implantable medical devices and their biocompatibility should be further verified. Research on utilizing gallium-based liquid metals as bioimplants is a very recent development, therefore, FDA-approved implants based on gallium or indium metal have not yet produced. While we agree that there can be differences in properties between FDA-approved gallium/indium salts and the respective metals, salts generally exhibit enhanced biological interactions due to their ability to readily form metal ions than the metal itself. Consequently, we have proposed the biocompatibility of gallium and indium salts as a positive indicator for the biocompatibility of the corresponding metals, as well as examples of the use of liquid metals in biomedical devices with references.

In our manuscript, we have pioneered the use of liquid metal (EGaIn) as a material for brain probes and their subsidiary electronics. Cytotoxicity tests (Supplementary Fig. 22) and in-vivo toxicity tests (Supplementary Fig. 23, 24) were conducted, demonstrating a certain level of biocompatibility. Also, continuous long-term neural recording for 8 months further substantiated the biocompatibility of the liquid metal-based brain implant.

[**Supplementary Fig. 22.** MTT and Calcein assays of SH-SY5Y cells cultured in media pre-contained with the pristine EGaIn and PtB/EGaIn samples for 7 days. Scale bars, 50 μ m.]

[**Supplementary Fig. 23.** Fluorescence micrographs of a horizontal section of the mouse brain 8 weeks after implantation with our soft neural probe by capillary-assisted injection, compared to the control sample with no probe injection. Neurons, astrocytes, and microglia were stained with NeuN (green), GFAP (blue), and Iba1 (red), respectively. Scale bars, 50 μm .]

[**Supplementary Fig. 24.** Histology images of the hematoxylin and eosin (H&E) stained scalp skin of a mouse 6 weeks after the cranial circuit formation. Scale bars, 2 mm (left), 200 μm (right).]

The mechanical-modulus similarity to tissues and the low toxicity of gallium-based liquid metals have attracted attention for soft bioimplants. For further advancement in their bioelectronic applications, we agree that it is essential to conduct an analysis of the acute and chronic systemic toxicity of gallium, indium, and their alloys, including long-term clinical observation (throughout the life cycle) and dose-related studies. We have included this discussion in our revised manuscript.

Revised manuscript (page 24):

Gallium is known to have minimal cytotoxicity and has negligible solubility in water (except as a salt). The United States Food and Drug Administration (FDA) has approved the use of gallium salts, such as gallium citrate, as medical imaging agents for imaging cancer and inflammation via intravenous injection³⁹. Indium is also negligibly soluble in water, and the FDA has approved indium-based materials as clinical diagnostic agents for detecting endocrine tumors and infection⁴⁰. **Generally, metal salts exhibit enhanced biological interactions, so the biocompatibility of gallium and indium salts could be represented as a positive indicator for the biocompatibility of those metals.**

Recently, in-vivo toxicity studies showed that EGaIn is non-toxic based on histological and blood panel indications at high oral administration⁴¹ or surgical implantation⁴². Therefore, liquid metals have allowed the development of a range of biomedical devices, including biosensors^{43,44}, blood vessel⁴⁸, and implantable devices⁴².

Revised manuscript (page 24):

Future considerations for the potential translation of our neural recording system and on-tissue printing method to biomedical applications include i) ~~evaluation of systemic toxicity~~~~verification of the biocompatibility~~ of liquid metals and devices for ~~acute and chronic responses~~~~long-term implantation~~, and ii) comprehensive studies on biocompatibility including ~~dose-related studies and long-term clinical observation (throughout the life cycle)~~ ~~human clinical trials~~.

Comment 2: “Concerns about the limited recording capability of the devices. Each device has a relatively small number of channels (12 channels), which restricts their ability to record from only a small number of neurons in each animal. Notably, in 2017, Fu et al. have demonstrated stable long-term recording using 128-channel flexible mesh device in each animal (PNAS, 114, E10046-E10055, 2017). More recently, used flexible probes and achieved month-long stable recordings from approximately 1000 neurons in each animal (Nature Biomedical Engineering, doi.org/10.1038/s41551-022-00941-y). These examples indicate the need for the authors to consider how to enhance the recording capacity of their devices in order to compete with current advancements in the field.”

Response to Comment 2:

We agree with the advancements in the field demonstrated by a large number of channels to improve spatial resolutions for an understanding of neural activity. We recognize the importance of keeping pace with current advancements in the field as well as feature the advantage of our systems.

To apply our strategy to a greater number of channels, it is required to achieve higher printing speed in forming interconnections and circuits with high yield, along with the development of multi-channel probes from our soft neural probes. With these advancements, our printing-based approach can create customizable neural interface systems tailored to various shapes and arrangements of multiple and multi-channeled neural probes. We added this discussion in the revised manuscript.

Revised manuscript (page 25):

Through the further development of multi-channeled soft neural probes and their tailored circuit printing, our approach can acquire neural signals with high spatial resolutions for broad areas.

Revised manuscript (page 26):

However, short procedure time and high fabrication yield are preferred for human translation. Printing on the human brain, having a much larger surface area, requires improvements in printing speed compared to the small animal model cases. Also, it should provide high yield while forming larger circuits. We believe that such issues can be addressed through real-time surface profiling of the cranium by laser or depth camera⁵¹, the selection of passivation layers with enhanced adhesion to liquid metals⁵², and the modification of liquid metals’ rheological properties¹³ to allow high-yield, fast conformal printing.

Comment 3: “Concerns about the potential impacts of the current in-surgery direct-writing approach. This method comes with several drawbacks, including reliability issues and an increase in surgical time. It’s important to note that all fabrication methods have yield issues, but prefabricated devices have the advantage of allowing for the elimination of defective products before implant surgery, reducing risks to the patient. In contrast, the use of direct written interconnection devices significantly raises the risk of failed implanted devices, which is a considerable concern. Additionally, the increased surgical time associated with this approach can lead to increased intraoperative risks. These aspects warrant careful consideration and further evaluation of the practical feasibility and value of the in-surgery direct-writing method.”

Response to Comment 3:

We agree that short procedure time and high fabrication yield are preferred for human translation. The printing on the larger animals or human brain, having a much larger surface area, requires an improvement in printing speed compared to that for small animal models. Also, it should provide high yield while forming larger circuits. We believe that such issues can be addressed through real-time surface profiling of the cranium by laser or depth camera, the selection of passivation layers with enhanced adhesion to liquid metals, and the modification of liquid metals’ rheological properties to allow high-yield, fast conformal printing. We added this discussion in the revised manuscript.

Revised manuscript (page 26):

However, short procedure time and high fabrication yield are preferred for human translation. Printing on the human brain, having a much larger surface area, requires improvements in printing speed compared to the small animal model cases. Also, it should provide high yield while forming larger circuits. We believe that such issues can be addressed through real-time surface profiling of the cranium by laser or depth camera⁵¹, the selection of passivation layers with enhanced adhesion to liquid metals⁵², and the modification of liquid metals' rheological properties¹³ to allow high-yield, fast conformal printing.

Revised manuscript (page 39):

13. Park, Y.-G. et al. Three-Dimensional, High-Resolution Printing of Carbon Nanotube/Liquid Metal Composites with Mechanical and Electrical Reinforcement. *Nano Lett.* 19, 4866–4872 (2019).

Revised manuscript (page 42):

51. Jafari, B. H. & Gans, N. Surface Parameterization and Trajectory Generation on Regular Surfaces With Application in Robot-Guided Deposition Printing. *IEEE Robot. Autom. Lett.* 5, 6113–6120 (2020).

52. Zhao, R., Guo, R., Xu, X. & Liu, J. A Fast and Cost-Effective Transfer Printing of Liquid Metal Inks for Three-Dimensional Wiring in Flexible Electronics. *ACS Appl. Mater. Interfaces* 12, 36723–36730 (2020).